



# Microphysical fingerprints in anvil cloud albedo

Declan L. Finney[1,2], Alan M. Blyth[2,1], Paul R. Field[3,1], Martin I. Daily[1], Benjamin J. Murray[1], Mengyu Sun[4], Paul J. Connolly[4], Zhiqiang Cui[1,2], and Steven Böing[1]

[1]Institute for Climate and Atmospheric Science, School of Earth and Environment, University of Leeds, Leeds, UK
[2]National Centre for Atmospheric Science, Leeds, UK
[3]Met Office, Exeter, UK
[4]Centre for Atmospheric Science, Department of Earth and Environmental Sciences, University of Manchester, Manchester, UK

**Correspondence:** Declan L. Finney (d.l.finney@leeds.ac.uk)

**Abstract.** Improved understanding of anvil cloud radiative effect and feedback is critical for reducing uncertainty in climate projections, with recent research highlighting cloud microphysics and anvil albedo as requiring further investigation. In this study, we use nine observation-informed model experiments to simulate a 24-day period from the Deep Convective Microphysics Experiment (DCMEX), with our analysis quantifying the influence of cloud microphysics on high cloud albedo. We
find that increasing cloud droplet number (2x) or ice nucleating particles (INP) ($\sim$10x), within the range of observed variability, significantly increased high cloud albedo by 1-3 % (p-value<0.05). To isolate the microphysical drivers of albedo changes, we introduce fingerprint metrics based on an ice water path (IWP) threshold, distinguishing between thick and thin high clouds. We find that increased droplet number enhances albedo in both thick and thin clouds, while higher INP concentrations primarily affect thick cloud albedo. These fingerprints offer a novel approach for elucidating causes of variability in high cloud albedo in
both models and observations. Future work should explore how the fingerprints translate across different high cloud regimes and global climate context. Beyond direct microphysical influences, we also identify strong correlations between albedo and large-scale environmental factors such as relative humidity, thereby motivating future investigation of anvil albedo feedback using cloud controlling factor analysis. Our study highlights both the large-scale environment and microphysical processes as important for accurate prediction of cloud radiative effects and feedbacks in climate models.

## 1 Introduction

Understanding the effect of clouds on radiation has been a long-running focus of atmospheric research, and the topic is particularly active with regard to the interaction of clouds and climate change, also known as cloud feedbacks. In a comprehensive review of cloud feedbacks in 2020 (Sherwood et al., 2020), it was proposed that high cloud area feedback was one of the largest uncertainties and could have a large negative value. However, in 2024, three new lines of analysis of tropical high cloud feed-
back (or *anvil* cloud feedback) have rejected the potentially strong negative feedback included as plausible by Sherwood et al. (2020), instead concluding that a near-zero area feedback is likely (Raghuraman et al., 2024; McKim et al., 2024; Sokol et al., 2024). Most importantly, it has been identified that the decomposition of the "tropical anvil cloud area" feedback by Sherwood et al. (2020), using Williams and Pierrehumbert (2017), conflated effects of area and optical depth changes (McKim et al.,



2024). Optical depth feedback is likely dominated by changes in the shortwave, i.e. albedo, and McKim et al. (2024) concludes
that the anvil albedo feedback remains poorly constrained and could represent a large uncertainty in total anvil cloud feedback.
They suggest that a 1-2 % $K^{-1}$ change in high cloud albedo could result in a feedback as large as the area feedback proposed
by Sherwood et al. (2020). With current knowledge it is challenging to estimate a plausible range of high cloud albedo change,
let alone the most likely albedo change.

Understanding the drivers of variability in high cloud albedo, or optical depth, when considered directly, is inherently a
microphysical problem. Changes in albedo occur as a result of changes in particle phase, mass, number or shape for a given
cloud area. McKim et al. (2024) and Sokol et al. (2024) both identify microphysics as a key uncertainty in their feedback
estimates that requires further work, while Raghuraman et al. (2024) highlights that aerosol indirect effects are intertwined
with their feedback estimate.

There is a substantial literature exploring the interaction of aerosol and deep convection (e.g. Grabowski and Morrison,
2016; Heikenfeld et al., 2019; Igel and van den Heever, 2021; Barthlott et al., 2022; Varble et al., 2023). The subset of
literature which also considers the anvil radiative effect is relatively small, but there are a number of important studies to
draw from. Gasparini et al. (2023) provide an opinion piece on linking tropical high cloud microphysics and climate impacts.
They advocate for a holistic approach going forward, with a need for studies that propagate ice microphysical uncertainties
to macroscale phenomena such as radiative effects. Fan et al. (2013) provide an existing example of such an approach by
modelling clean and polluted development of deep convective clouds in three locations including the tropical west Pacific
and Southern Great Plains of the USA. They find that by increasing cloud condensation nuclei, there is a decrease in net
cloud radiative effect (CRE) top-of-atmosphere, as a result of increased anvil cloud area and albedo. They found that this
change in anvil cloud occurred due to smaller and more numerous ice crystals staying aloft in the high cloud. Hawker et al.
(2021b) explore a wider range of microphysical controls on cloud radiative effect, by investigating primary ice formation by
ice nucleating particles (INP) and secondary ice production (SIP). Their results suggest INP play an important role in anvil
cloud radiative effect, even in the presence of SIP.

In addition to direct microphysical influences, Fan et al. (2013) highlight the role of environmental conditions such as wind
shear and humidity on modifying the radiative effect of the high condensation nuclei concentration. Other works have also
found important radiative influence of various environmental conditions, such as: shear (Lin and Mapes, 2004), atmospheric
circulation and humidity (Gasparini et al., 2022), upper tropospheric winds and stability (Wilson Kemsley et al., 2024), and
timing of convection with the diurnal cycle (Jones et al., 2024). Meanwhile, Miltenberger et al. (2018b) highlight that an
ensemble of meteorological conditions is needed to significantly determine effects of aerosol on cloud properties. Together
these highlight the need to consider interactions of thermodynamic, dynamic, microphysical and timing of convection when
investigating the influence of anvil cloud on radiation.
The Deep Convective Microphysics Experiment (DCMEX) project aims to improve our understanding of microphysics and
its impact on anvil CRE. The project team undertook a field campaign at the Magdalena Mountains, New Mexico, USA in
summer 2022 which used aircraft, radar, and a range of other instruments to study the aerosol, microphysics, thermodynamics
and dynamics during the deep convective growth stage (Finney et al., 2024). The target clouds occurred over the same mountain,





so, through a common elevated trigger for convection, offer a *pseudo-control* on the convective forcing. Given that similar
forcing of different cases, variation in clouds due to other environmental factors can be more readily distinguished. The 17
flight days observing deep convective cloud formation sampled a range of environmental conditions including variation in
humidity and wind shear. A number of studies in DCMEX have investigated the aerosol and microphysical processes observed
during the campaign (Daily et al., 2025; Evans et al., 2025). DCMEX builds on a wealth of information gathered in previous
projects over the Magdalena Mountains (e.g. Raymond and Wilkening, 1982, 1985; Raymond and Blyth, 1989).

In this study, we define a set of microphysics experiments, informed by the DCMEX observation data, to drive convection-
permitting simulations. These experiments vary cloud droplet number, primary and secondary ice formation, and the sub-grid
representation of mixed-phase cloud. Given the large number of DCMEX cases with similarly-forced deep convective clouds,
forming over the same mountain at approximately the same local time under a range of environmental conditions, we are able
to statistically assess the impact of microphysics and the environmental thermodynamic and dynamical conditions on the anvil
cloud albedo and CRE.

## 2 Model and observation data description

A suite of case study simulations have been generated for the DCMEX campaign period. These have been evaluated against
satellite-based, top-of-atmosphere radiation products, and then analysed to understand the drivers of albedo and radiative effect
variation.

### 2.1 The UM-CASIM model

The convection-permitting model used in this study is the Unified Model (UM) with the Cloud AeroSol Interacting Micro-
physics (CASIM) module (Field et al., 2023).

The Unified Model version used is the Regional Atmosphere-Land 3 (RAL3.2). This is based upon RAL2, as described
by Bush et al. (2023), with changes from the bimodal cloud fraction scheme (Van Weverberg et al., 2021) and CASIM cloud
microphysics scheme (Field et al., 2023). A number of recent studies have used the RAL3.2 model version to study clouds of
varying types (Weverberg et al., 2023, 2024; Maybee et al., 2024; Huang et al., 2025).

CASIM is a multi-moment, bulk cloud microphysics scheme representing cloud using five hydrometeor species: cloud liquid,
rain, ice crystals, snow, and graupel. Here we use a 2-moment configuration where each hydrometeor species is modelled using
prognostic mass mixing ratio and number concentration.

Formation of cloud liquid and ice crystal hydrometeors in the model can be simulated interactively based on aerosol, or
using a more prescriptive approach of fixed parameters and functions which decouple the cloud hydrometeor species from
the simulated aerosol (Field et al., 2023, Appendix A.9). Secondary ice formation in the model is limited to the most widely-
studied process, the Hallett-Mossop process, producing a number of ice splinters per kilogram of rimed liquid according to a
triangular function between -2.5°C and -7.5°C, peaking at -5°C (Field et al., 2023, Appendix A.10).





UM-CASIM simulates cloud liquid and ice fractions diagnostically. When both of these fractions are less than one in a given grid cell, an assumption needs to be made of the degree of overlap of the liquid and ice cloud at the subgrid level in order to determine rates of mixed-phase processes (Field et al., 2023, Appendix A.6.1). This is prescribed with a constant mixed-phase overlap fraction parameter where a value of 0 minimises overlap and value of 1 maximises overlap.

CASIM has been successfully used in a number of past studies to investigate the role of microphysics in convective cloud
(Miltenberger et al., 2018a, b, 2020; Hawker et al., 2021b). Work here explores a new location and different convective forcing, as well considering different sensitivities. A pertinent update in RAL3.2 for our purposes is that the radiation scheme now considers ice crystal and snow hydrometeor number as well as mass (Baran et al., 2025).

## 2.2 Model configuration

The specific model configuration for this study uses a 600 x 600 grid of ∼1.5 km x 1.5 km square cells (Figure 1a shaded
contour region) centred on the Magdalena Mountains (Figure 1b red contour) in New Mexico, where the DCMEX campaign was undertaken. Individual case simulations are made for each of 24 days within the period 16th July to 8th August 2024. For analysis, we select a 2° x 2° latitude-longitude domain around the centre (Figure 1a red box) in order to avoid analysing data affected by the boundary. This subdomain is dominated by thermally-forced, orographic, deep convective clouds and therefore provides a suitable domain with which to apply statistical analysis. The model column is simulated using 70 hybrid-height
vertical levels up to 40 km with higher resolution towards the boundary layer. The model time-step is 75 seconds, except for the radiation scheme which uses a time-step of 15 minutes. Given the importance of terrain to this study, we update the orography ancillary file to use the Shuttle Radar Topography Mission (SRTM) dataset.

The model is initialised using UK Met Office analysis fields for 0 UTC on the day of each case, simulates 36 hours, and the first 12 hours are discarded as spin-up. Local time in summer was Mountain Daylight Time, which is 6 hours behind UTC.
Where daily average values are presented, these are calculated over 12 to 12 UTC. Where time series of all cases are presented, these are stitched together from the individual case simulations.

The boundary conditions are provided by a 1-way coupled global model simulation with the UM with grid n640 (∼25 km spacing) and running the Global Atmosphere 8 - Global Land 9 (GA8/GL9) physics. The GA8/GL9 physics builds upon the GA7/GL7 physics described by Walters et al. (2019).

## 2.3 Model sensitivity experiments

The following sensitivity experiments target a number of key microphysical features that can be informed by aircraft observations and which may influence anvil CRE. Cloud droplet number concentration has been widely studied, and previous results have suggested the size and number of droplets are important regarding convective invigoration, ice hydrometeor composition and CRE (e.g Fan et al., 2013; Miltenberger et al., 2018b; Igel and van den Heever, 2021). There is evidence that primary and
secondary ice formation in deep convective clouds could affect radiation (Hawker et al., 2021b). We also explore the model sub-grid representation of the mixed-phase region (i.e. how liquid and ice cloud is assumed to overlap within grid-cells), with observational constraint of this parameter now possible due to a recent study (Evans et al., 2025). Secondary ice production via





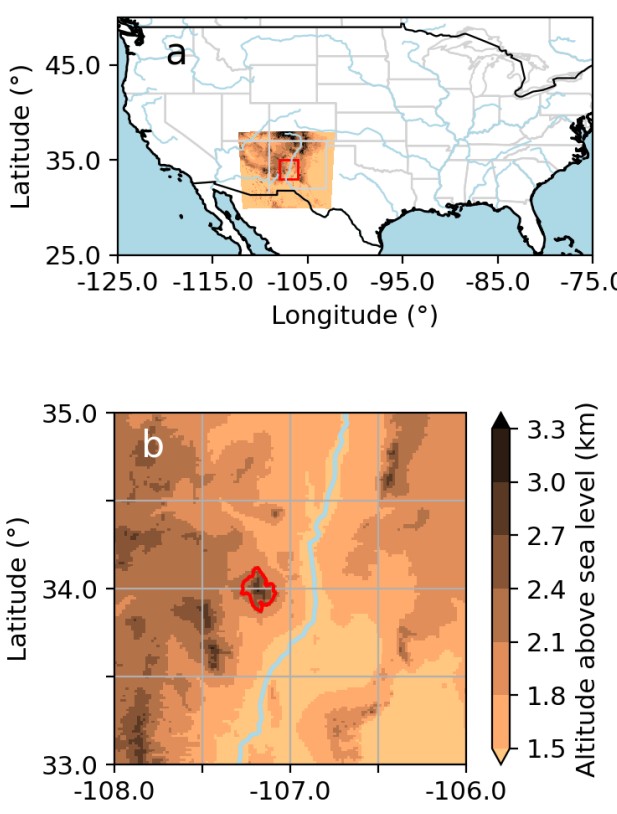

**Figure 1.** Model and analysis domains. Panel a shows, in the context of North America, the full model domain as the region with model orography shaded in copper colours. State and country borders are marked with grey and black lines, respectively. For analysis a 2° x 2° sub-domain is used (red box in panel a), and a zoomed-in view of this domain is shown in panel b. In panel b, the 2250 m orography red contour around the Magdalena mountains is included to highlight where the DCMEX field campaign was focused. Rivers are marked in light blue, with the Rio Grande apparent running north to south through panel b.



the Hallett-Mossop process depends on riming and therefore could be affected by the sub-grid representation of mixed-phase cloud.

Cloud droplet number concentration is largely determined by aerosol emission and transport. However, to constrain our experiments to focus on the in-cloud processes, we prescribe a fixed profile of cloud droplet number concentration. This is defined by a constant droplet number per kilogram of air, a height above ground level at which droplet number exponentially decays, and the exponential decay rate. Cloud droplet number concentration was measured with the Cloud Droplet Probe 2 on the research aircraft during DCMEX flights, as described by Finney et al. (2024). Profiles of individual flights are highly

variable and dependent on the case sampling decisions and representativeness. Therefore, results are statistically analysed across all flights to get a representative profile and estimate of variability across cases in the DCMEX campaign. Figure 2a shows the mean and deciles of in-cloud cloud droplet number concentration once averaged over 1.5 km straight-and-level flight legs and binned in 500 m segments of height above ground level (AGL). Robust statistics (sample number > 20) were collected up to 6 km AGL. It is clear that droplet numbers undergo very little reduction with height. The gradual reduction of

model number concentration with height in Figure 2a is a result of decreasing pressure for a fixed number mixing ratio, and is similar to the reduction seen in the observations. Given the absence of evidence above 6 km AGL, we assume the default model exponential decay of rate of 0.0004 m$^{-1}$ from this point. We choose a control droplet number of 250 x 10$^6$ kg$^{-1}$, and sensitivities of 125 x 10$^6$ and 500 x 10$^6$ kg$^{-1}$ to explore the observed variability. When comparing these model profiles to the observations, it is apparent that the spread of the central 80-90% of measurements is well captured by the model sensitivity

experiments.

Primary ice production in the model can be nudged to the Cooper (1986) temperature-dependent distribution, as described by Field et al. (2023, Appendix A.9.1). However, this does not allow secondary ice production to occur freely, so for the control we simulate ice concentration to be at least the Cooper value for the simulated temperature, but it is taken as the prognostic ice crystal concentration if that is higher. During each flight of the DCMEX campaign, aircraft filter samples were collected

and analysed for concentrations of ice nucleating particles (INP). The filter samples were collected from a circuit around the base of the Madgalena mountains (Figure 2b) and at heights varying between 0.25 km to 5.5 km AGL. It can be seen that the Cooper curve (red line) sits within the spread of filter measurements. It is within an order of magnitude of the lower bound of the filter measurements. For much of the temperature range, particularly the warmer parts, the upper bound of measurements is at least an order of magnitude higher than the cooper curve. We explore this upper range with two sensitivity experiments

(orange lines) based on the work by Daily et al. (2025).

Daily et al. (2025) produced a new parametrisation for temperature-dependent primary ice formation based on the observed INP data. They found that a two-component fit to this INP data worked well, where the low temperature end of the spectra were well represented assuming mineral dust controlled nucleation, whereas the higher temperature end is thought to be related to biological ice nucleating entities. To describe the low temperature component, we use a parameterisation derived for K-feldspar

(Harrison et al., 2019) and a power law is used to represent the strong curvature in the INP spectra above about -25°C:

$$N_{INP} = A_{dust}F_{Kfsp}10^{n_s(T)} + ae^{(b(T_{max}-T)^c)} \tag{1}$$

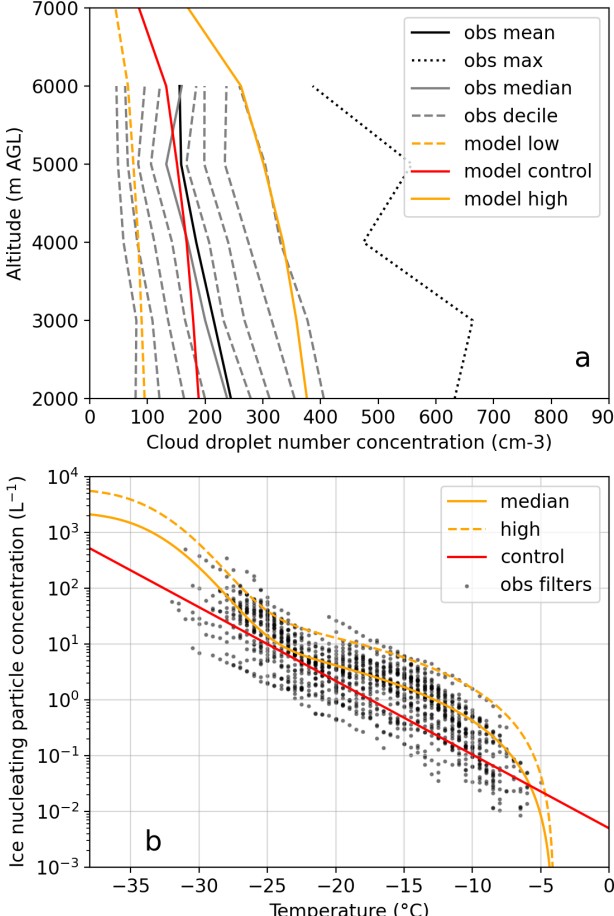

**Figure 2.** Observations of cloud droplets and ice nucleating particles informing model experiments. a) Statistical profiles of straight-and-level, altitude-binned aircraft observations of cloud droplet number concentration (black) against the model control profile (red) and two sensitivity experiment profiles (orange) described in the main text. b) Filter-based measurement of INPs from aerosol runs when aerosol $>= 100$ cm$^{-3}$ (black), the existing Cooper (1986) parametrisation (red), and two new parametrisations (orange) developed by Daily et al. (2025) from the data presented here.



where $A_{dust}$ is the surface area of aerosol mineral dust present, $F_{Kfsp}$ is the proportion of the mineral dust that is composed of K-feldspar, and $n_s(T)$ is the temperature dependant ice-nucleating site density for K-feldspar from Harrison et al. (2019). For the median fit (solid orange line, Figure 2b), $A_{dust}$ = 4.94 $\mu m^2\ cm^{-3}$, based on Scanning Electron Microscope with Energy Dispersive Spectroscopy (SEM-EDS) analysis of filters from the DCMEX campaign, and $F_{Kfsp}$ = 0.05, consistent with a typical K-feldspar mineral dust content. $T$ is cloud temperature (°C), and $T_{max}$ = -4 °C. Daily et al. (2025) determined a median set of parameter values fitted to filter INP data from aerosol runs when aerosol $>=100\ cm^{-3}$. The parameter values are: $a$ = -1.93 x $10^{-22}\ L^{-1}$, $b$ = 45.25 $°C^{-1}$ and $c$ = 0.046. A high INP sensitivity set of parameters was determined which fits the upper bound of measurements (dashed orange line, Figure 2b): $a$ = -1.93 x $10^{-22}\ L^{-1}$, $b$ = 46.9 $°C^{-1}$, $c$ = 0.041 in Equation 1, and mineral surface area of 13.01 $\mu m^2\ cm^{-3}$ in the Harrison et al. (2019) parametrisation.

These DCMEX derived INP concentration spectra along with the Cooper curve for primary ice number concentration are shown with the INP filter measurements in Figure 2b. The DCMEX parametrisation reproduces the complex shape of the INP spectra observed during DCMEX, unlike the Cooper curve. The Cooper curve generally sits at the lower end of the INP measurements, whereas the median parametrisation predicts higher primary ice concentrations than the Cooper curve at temperatures less than approximately -6 °C. The greatest differences between the control (i.e. Cooper curve) and median curves are about one order of magnitude. The high DCMEX INP parametrisation produces higher ice concentrations than the median curve at all temperatures, with the greatest differences between the control and high curve being about a factor of 15.

The default heterogeneous ice formation in UM-CASIM occurs only when temperatures are colder than a set value. In the control simulation this is -8 °C, but the INP measurements indicate that ice formation may be possible at warmer temperatures. We explore the implications of this with a sensitivity experiment allowing heterogeneous ice production at temperatures of -5 °C or colder.

Observational estimation of Secondary Ice Production (SIP) is complex and work is ongoing to quantify the campaign SIP processes and rates. In the default UM-CASIM, only the Hallett-Mossop SIP is active, as described by Field et al. (2023, Appendix A.10). The control SIP rate is 350 x $10^6$ ice splinters per kg of rimed liquid at -5 °C. We explore the sensitivity of anvil radiative properties to the SIP rate by setting the rate to zero, and by doubling it.

As introduced in Section 2.2, the simulation of SIP and other cloud processes depends on the subgrid representation of mixed-phase regions. Recent work with the DCMEX data and other aircraft campaign datasets has provided the first observational estimate of this parameter (Evans et al., 2025), estimating a value of 0.85 for deep convective cloud. We use this value in our control simulation, and also generate a sensitivity experiment using a value of 0.5, which has typically been used as the default in CASIM.

The set of simulations described above are summarised in Table 1.

## 2.4 Satellite and reanalysis data

The primary data source for radiation evaluation used is the Clouds and the Earth's Radiant Energy System (CERES). Two products are used: the hourly CERES Synoptic product (SYN1deg) (Doelling et al., 2016) and the daily CERES Flux By Cloud Type product (FBCT) (Sun et al., 2022). CERES-SYN1deg combines CERES and geostationary satellite top-of-atmosphere



(TOA) radiative fluxes to generate a 1° x 1° hourly gridded dataset. Moderate Resolution Imaging Spectroradiometer (MODIS) and geostationary satellite instruments are used for cloud properties. CERES-FBCT uses the MODIS radiances within a CERES footprint from daytime overpasses of the Terra and Aqua satellites, decomposed by cloud type, to estimate the cloud-type broadband fluxes. MODIS-derived broadband fluxes were found to be within 1 % and 2.5 % of the CERES-observed footprint fluxes for longwave (LW) and shortwave (SW), respectively. The data are available as daily averaged fluxes stratified by cloud top pressure and optical depth on a 1° x 1° grid. For this analysis, model and CERES-SYN data are averaged to 12-12 UTC daily means to compare over the DCMEX cases. For equivalent radiative fluxes, CERES-FBCT is interpolated from its daily means centred on 12 UTC, to values representative of means centred on the following 0 UTC.

We also used narrowband radiances per unit wavelength from channels 2 (visible) and 13 (infrared) of the GOES-16 geostationary Advanced Baseline Imager with central detection wavelengths of 0.64 and 10.3 $\mu m$, respectively. These provide a qualitative comparison to the model outgoing SW and LW fluxes on fine spatial and temporal scales.

In order to explore the environmental drivers of albedo variability, we use ERA5 reanalysis data (Hersbach et al., 2020). In particular, the following variables are used: Convective Available Potential Energy (CAPE), relative humidity on pressure levels, and eastward and northward winds on pressure levels. The CAPE variable provided within the ERA5 data is the maximum CAPE of parcels initialised from model levels below 350 hPa. The closest equivalent to this definition in UM-CASIM model diagnostics is maximum CAPE of parcels initialised from model levels below 500 hPa.

All datasets from section 2.4 are spatially averaged over the 2° x 2° domain in Figure 1.

## 3   Analysis methods

For geostationary satellite evaluation, raw GOES data was obtained using GOES-2-go python package (Blaylock, 2023), a smaller region around the southwest USA was retained and regridded to a regular latitude-longitude grid with spacing equal to the mean latitudinal spacing in the raw data. For this study, we did not correct for the parallax effect. In our domain, parallax offset of the satellite cloud images from their true position is roughly proportional to the cloud top height. Positioning errors of up to around 10–15 km will occur but are not important for our analysis as the evaluation is qualitative.

For analysis of the UM-CASIM cloud properties and radiation, it is necessary to identify the cloud top model level and its properties. Cloud top model level was calculated using the following method:

1. Total cloud fraction and mass mixing ratios of all hydrometeor species are archived instantaneously on the hour for the full 3D grid.

2. For each hour, the total cloud mass mixing ratio is calculated.

3. Grid cells are classified as cloudy if their cloud mass mixing ratio is greater than $1 \times 10^{-6}$ kg kg$^{-1}$ and their total cloud fraction equals 1.

4. For each column, the highest model level meeting the above condition is identified. The level number as well as the level's properties, such as pressure, are retained.



Consistent with the International Satellite Cloud Climatology Project and CERES-FBCT threshold, we define high cloud as that with cloud top pressure less than or equal to 440 hPa.

In analysis where variables are on different grids or archived at different temporal frequencies, they are bi-linearly regridded and/or linearly interpolated in time to a common grid and times. If a cloud mask is regridded or interpolated then it is used as a weighting in order to account for decimal values.

Robust evaluation between the model and CERES-FBCT requires some care. CERES-FBCT radiation and cloud fraction estimates are based on daytime overpasses of AQUA and TERRA satellites. For our cases and analysis domain, these overpasses

lie between 16 and 22 UTC. In order to compare similar cloud variability from the model, we select this time range to average the model radiative fields. However, within the CERES-FBCT product, a diurnal albedo model is applied to adjust the satellite overpass measurements to an estimate of daily radiative fluxes, assuming cloud and atmospheric conditions remain constant over the day. A consequence is that the CERES-FBCT SW fluxes are much lower than one finds just by averaging the model 16-22 UTC fluxes. We approximately account for this by adjusting the UM-CASIM outgoing SW fluxes by finding, on a daily

basis, the ratio between the CERES-FBCT daily and the UM-CASIM 16-22 UTC incoming SW fluxes. This then is used to scale the model flux estimate when comparing to CERES-FBCT. This approach is only used for evaluation purposes of UM-CASIM domain-mean, high cloud SW CRE comparisons to CERES-FBCT in Figure 4f and Table 2 and will be referred to the "Approximate domain mean, high cloud daily SW CRE". In Figures 5c and 6c, the focus is only on UM-CASIM data so we revert to the unadjusted value of domain mean, high cloud daily SW CRE which is only available from simulations as the

radiative fluxes are not measured by satellite at all times of day. Unlike the radiation, the CERES-FBCT high cloud and albedo products include no diurnal adjustment. This means these variables can be directly compared to the UM-CASIM mean over the 16-22 UTC period. For this study, cloud radiative effect is defined as clear-sky, top-of-atmosphere, outgoing radiation minus the all-sky, top-of-atmosphere, outgoing radiation.

Percentage anomalies of each microphysics experiment relative to the control are used to understand the effect of micro-

physics. The significance of these results is tested using a one-sample, two-tailed t-test with a null hypothesis of a mean of zero, and requiring a p-value less than 0.05.

Multiple linear regression is used to consider dynamic and thermodynamic influences on case variability of high cloud radiative effect. This is performed over the whole simulated period for the DCMEX model analysis domain (Figure 1a red box). Many regression models are produced using different combinations of the three chosen variables. Adjusted $R^2$ is used

to assess the relative explanatory power of regressions, given the different numbers of explanatory variables. Significance is noted where regression fits have a p-value less than 0.05.

## 4    Model evaluation

In Figure 3, the cloud morphology, as viewed by geostationary satellite and from the model control experiment, is shown for 4 example cases for 18:30 and 21:30 UTC (12:30 and 15:30 local). Model data are hourly average outgoing shortwave radiation,



whilst the satellite data are the average radiance per wavelength from all 5-minute scenes available for the same hour from a
GOES-16 visible channel. These two variables are not quantitively equivalent, but should be qualitatively similar.

Figure 3a-d shows data for the 17th July. The satellite images show that there was no cloud formed during this case. The
model produces a small amount of cloud but satisfactorily maintains largely clear skies.

Figure 3e-h shows data for 19th July case, when the model successfully simulates the orographically triggered cumulus at

18:30 UTC, which becomes more widespread at 21:30 but maintains a speckled pattern of isolated convection. The model
correctly maintains an absence of cloud along the Rio Grande which runs from top to bottom through the middle of the domain
(see Figure 1).

Figure 3i-l, for the 27th July case, provides good evidence that the model is able to simulate the varying nature of the anvil
cloud. It shows a very different day of convective cloud development compared to the 19th July case. The model captures

important aspects of the variation. Notably, at 18:30 UTC both satellite and model show fairly narrow streaks from the bottom-
right to top-left direction, with smoother and more widespread streaks from the top-right to bottom-left direction. This was
observed during the campaign when ~90° wind shear between lower and upper levels resulted in differing directions of cloud
advection as the clouds grew (Finney et al., 2023, timelapse footage). At 21:30 UTC both satellite and model show widespread
cloud cover, notably different to the cloud development on the 19th July (Figure 3e-h).

Figure 3m-p presents an example of a failed case in the model simulation. Our chosen setup of initialising at 0 UTC can
result in previous-day or advected cloud present at the beginning of the day not being well represented. This was the case
on the 4th August. It is apparent from Figure 3o and p that the model-generated high cloud was too thick and widespread
compared to observations. Supplementary Figures S8 g-h show that the model did not capture the observed widespread high
cloud coverage at 12:30 UTC. This would have affected the development of the overnight atmospheric conditions, and the early

morning solar irradiance reaching the surface. The model did not have a realistic night-time reduction of outgoing longwave
radiation that would have led to a warmer atmosphere, nor the morning reduction in incoming shortwave that would have led to
a cooler surface. Therefore, the model atmospheric profile would have been unrealistically unstable and more likely to generate
convective activity resulting in the widespread anvil cloud. Given our focus is on variability in the anvils we are less concerned
with simulating suppressed convection days (which is more a function of the triggering of convection, than cloud physics). We,

therefore, choose to accept a small number failed cases such as the 4th August in order to maintain a consistent case setup.

Next we consider the performance of the model in simulating the broader cloud and radiation of the cases. The time series of
model radiation is compared to CERES observation estimates for a range of metrics in Figure 4. For evaluation of high cloud
albedo, the metric is calculated based on mean albedo over the cloudy area.

The model captures well the temporal variability in the domain all-sky outgoing SW and LW, with correlations of 0.74 and

0.84, respectively (Figures 4a-b). There is, however, a mean bias in the SW. Our concern in this study is more focused on how
SW CRE varies so this average bias does not prevent us drawing conclusions. The CERES-SYN product also allows us to
evaluate the timing of maximum high cloud. This metric will be sensitive to subtle differences, especially if several hours have
a similar high cloud area to the area at time of maximum. Therefore, comparison is limited to qualitatively determining that
both model and CERES show a number of cases with maximum high cloud during the evening (blue band), before or close







**Figure 3.** Qualitative comparison of cloud morphology for a subset of cases. Showing observations by GOES-16 ABI channel 2 ("obs") and outgoing top-of-atmosphere SW flux simulated by the UM-CASIM ("model"). Each row is one of four example cases. The left two columns are the model and observations for the 18:30 UTC time, while the right two columns are the equivalent for the 21:30 time. Model-obs pairs are bounded by a black box. Brown contours are 2250 m orography. The thumbnail domain is 106-108 W, 33-35 N, as shown in Figure 1. Equivalent plots for all other simulated cases are provided in Supplementary Figures S1-5, and plots for a LW/infrared at 12:30 UTC are given in Supplementary Figures S6-8.





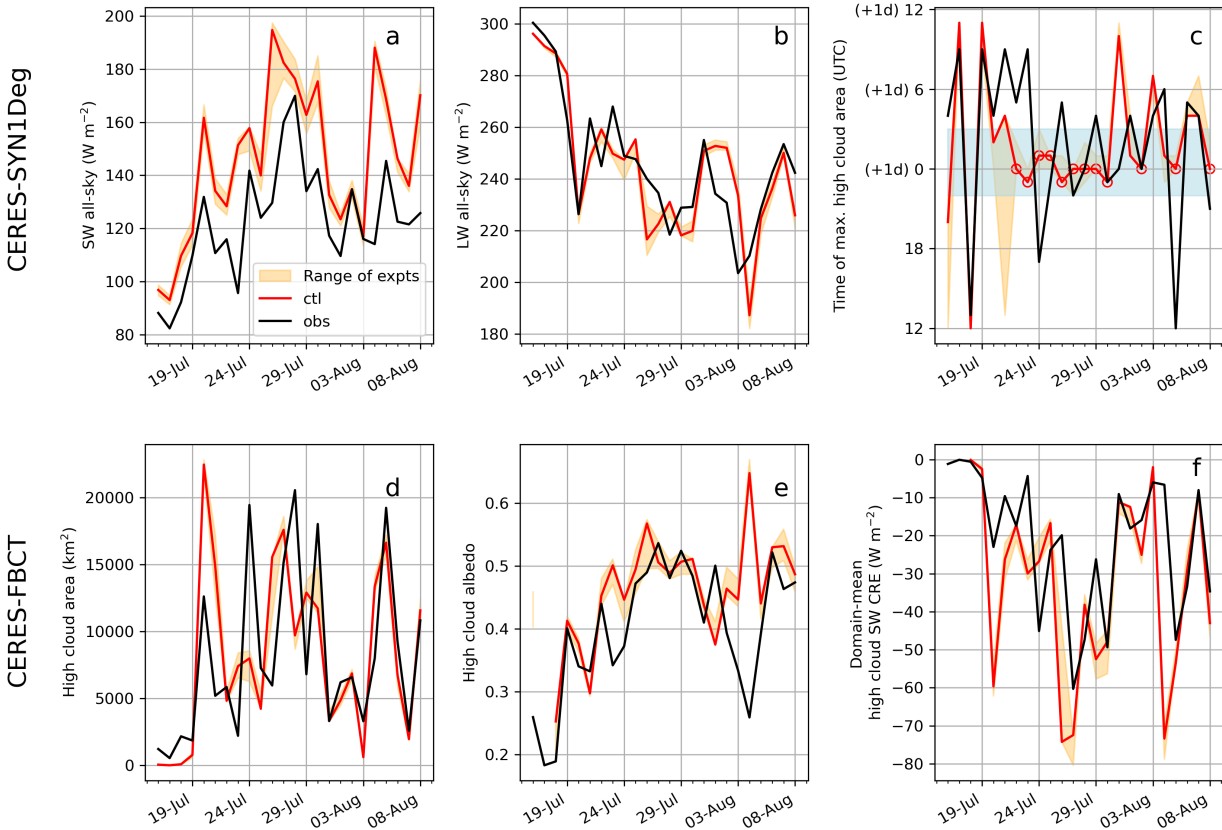

**Figure 4.** Comparison of radiation and cloud metrics daily time series for observations (black), control simulation (red) and model experiment spread (orange shading). Time series are stitched together from individual case simulations of the DCMEX 2022 campaign cases. Daily statistics for evaluation with CERES-SYN1Deg are calculated from 12 UTC to 12 UTC, and plotted on the start day. Panels a-c provide all-sky, top-of-atmosphere outgoing shortwave and longwave, along with time of maximum high cloud area. Panels d-f evaluate the model against CERES-FBCT 16-22 UTC mean high cloud area and albedo, and approximate domain mean, high-cloud daily SW CRE (as described in Section 3). The blue band on panel c highlights the range 22-03 UTC, with open red circles highlighting the cases where all experiments with UM-CASIM (orange shading) had peak high cloud during that blue shaded time period. The 4th August is not circled, despite meeting the condition, because the simulated high cloud albedo for that case is anomalous. In panel c y-axis, "+1d" denotes the following day from when a simulation is initialised. For reference, when interpreting the high cloud area, the full analysis domain is ~41000 km$^2$.





to sunset (∼02:30 UTC). Times of maximum high cloud are more erratic towards the beginning of the period, but these are
generally periods of lower high cloud coverage (Figure 4d) and therefore of less interest.

The model simulates a maximum in high cloud area across all experiments between 22-03 UTC (blue shading) for 13 cases.
We disregard the simulated 4th August case as this has already been identified as poorly simulated. The remaining 12 cases are
shown by red circles. These cases will be the focus of later sections as the consistency of peak time in high cloud area provides
some control for the influence of the diurnal solar cycle on the efficacy of albedo to impact CRE.

Looking at the remaining high cloud metrics (Figure 4 d-f), we are given confidence that the statistics of high cloud variability
are well captured by the model. Mean high cloud area is highly sensitive to the limited daily overpasses of AQUA and TERRA
satellites and cannot easily be directly compared with the model as convective timing is unlikely to perfectly align with reality
at the two daily overpass times. Yet, we are satisfied with the limited comparison here that the model provides a useful range of
variability in high cloud area (approximately 0–20000 km$^2$). The mean high cloud albedo is very well captured by the model
which, if the already identified 4th August outlier and NaN values are disregarded, captures the variation between albedo values
of approximately 0.25 to 0.55, with a correlation of 0.74. The domain-average, high cloud SW CRE (Figure 4f) encompasses
the combined effects of high cloud area and albedo. It can be seen that, as with other metrics, variability is broadly captured,
but there are a few cases with a negative bias, which explains cases with a larger bias in the all-sky outgoing SW.

The width of orange shading in all metrics, but particularly albedo and SW, is highly variable – with some days showing
wide experiment spread and others very little. Nevertheless, the larger case-to-case variability as opposed to experiment spread,
suggests the variability in high cloud area, albedo and radiative effect is dominated by environmental conditions as opposed
to the set of microphysical features we have tested here. Therefore, the next section explores the broad environmental controls
on these high cloud properties. Later sections then consider anomalies relative to the control in order to quantify the role of
microphysics.

## 5  Results

### 5.1  Association between environmental conditions and anvil cloud

As highlighted in Section 4 and Figure 4, there is large case-to-case variability in high cloud area, albedo and SW CRE and it
is larger than the spread for any individual case due to the perturbed microphysics experiments. This is likely, in part, the result
of large-scale environmental features having an influence. We therefore explore the relationship between three key daily mean
variables that capture broad variability in thermodynamics and dynamics in the campaign that may affect the anvil cloud:

- Relative humidity at 700 hPa (R700).

- Convective Available Potential Energy (CAPE) - most unstable CAPE is the CAPE metric used, as this is available from
  ERA5.

- Wind speed at 200 hPa (V200) - calculated from the daily mean northward and eastward winds, as the daily mean wind
  speed is not readily available from ERA5.



Broadly speaking, the motivation for these specific metrics is to explore the role of the cloud base moisture source (i.e. R700), the conduciveness of the thermodynamic profile for convection (i.e. CAPE), and the dynamical force dispersing the high cloud from the convective source (i.e. V200). We also explored a shear metric of the difference between V200 and the component of the wind speed at 700 hPa in the direction of the 200 hPa wind. Using this shear metric instead of V200 did not change which regressions were significant, and only slightly affected $R^2$ values. We therefore present the results of V200, with preference for the simpler of the two metrics.

Table 2 summarises the linear (and multi-linear) regression fits of each individual explanatory feature listed above (and some combinations of the features). For model regressions, the control simulation is used. Given the different numbers of explanatory variables used in the regressions, the adjusted $R^2$ value is used as the evaluation metric. In order to directly relate model regression results to observation-based regressions the same target variables as shown in Figure 4d-f (using 16-22 UTC data) have been used. These are regressed against daily mean explanatory variables. The use of 16-22 UTC averages for explanatory variables was tested. However, it was found that skill was generally worse for both the model and observations. This is possibly because the clouds themselves affect these variables but not in a way that adds skill for the period of study. Since daily means for predictor variables are convenient and make it more likely that they could be explored in larger-scale analysis, we have not investigated shorter timescale means any further.

Interestingly, given that many previous studies of high cloud have focused on explaining the area, the broadest conclusion from Table 2 is that high cloud albedo is the feature of cloud, in both model and observations, that is most predictable from simple environmental variables. In all cases, the model and observations have a feature that explains more variability in albedo (i.e. 44 and 67 % for model and obs, respectively) than they do for predicting high cloud area (13 and 36 % for model and obs). For all high cloud properties shown, a significant fraction of observed variability can be explained by these feature variables. An exception in the model, is that high cloud area is not well explained by the selected factors. Nevertheless, the finding that albedo is well predicted in both model and observation is encouraging for future cloud-controlling factor analysis to investigate the albedo (or optical depth) feedback.

When looking at the potential of the different feature variables to explain the high cloud properties, we see that both model and observations have significant linear fits to R700 and CAPE. Model and observations broadly agree that R700 is the strongest predictor for high cloud area and SW CRE, but the model shows a better fit with R700 for albedo, while the observations show a better fit with CAPE.

There is a significant fit in the model between V200 and SW CRE. Given the lack of correlation with high cloud albedo and area individually, the results suggest that model V200 is correlated with the co-variation of albedo and area, in order to be able to be significantly related to the variation in SW CRE. This could be reasonable to expect as, while the absolute values of area and albedo are likely set by R700, high wind speeds could be expected to add a secondary effect of spreading and thinning the cloud. The secondary nature of V200 is apparent from the lower correlation compared to R700, and the increased skill when combining the two feature variables in a multi-linear regression. We have already shown that winds did affect the observed cloud morphology (Figure 3, 19th and 27th cases) but it is not obvious why the regressions here do not show an observational relationship. It is possibly a limitation of the 16-22 UTC observation time period. A closer analysis of individual





cases is warranted to understand how shear interacts with other variables to modify high cloud. However, that is beyond the scope of this study.

Colours in Table 2 indicate positive/negative (red/blue) regression slopes, and show that higher R700 and CAPE are asso-
ciated with increased high cloud albedo and cloud area, and more negative SW CRE. These regressions are consistent with one another and are the same in both model and observations, where there are significant relationships in both. The model regressions suggest that when V200 is stronger then SW CRE increases (i.e. becomes less negative). This may be a result from a spreading and thinning of cloud particles or a greater rate of loss of cloud particles to sublimation in dry surrounding air.

This section has demonstrated that large-scale environmental variables can explain significant variability in albedo, high
cloud area and SW CRE, and that the model exhibits similar relationships to the observations. The next section will quantify the microphysical influence amidst this case-to-case variability.

## 5.2 Microphysical effects on anvil cloud

Figure 5 presents the model diurnal cycle in high cloud properties for the 12 days identified in Section 4 and Figure 4c. The aim of this figure is to assess the simulated diurnal-dependence of the microphysical effect across similar convective cases. The
black line shows the control simulation mean, and there is an orange line for the mean of each of the other experiments. The shading shows the spread across cases, and broadly speaking the control (ctl) simulation spread covers a majority of the widest spreads of any other experiment. We are not concerned with the particulars of each experiment here, just the broad differences of the experiments from the control.

The high cloud area (Figure 5a) at 12 UTC (6am local) has low, but non-zero, average values and a wide spread across cases.
Means of the different experiments are similar. The high cloud in this period results from the previous day's convection or advected cloud. The presence or not of such cloud will affect the environmental conditions that the day's storms form under, but in all cases the maximum high cloud area in these cases drops to low values by 17 UTC (11am local). One can see the albedo is dropping during this time (Figure 5b), likely as high cloud decays and the local heating after sunrise increases sublimation. During this early period, there is very little high cloud SW CRE (Figure 5c) as solar irradiance is low. The effect of preceding
conditions on convection is helpful to our analysis, as it provides an ensemble-like nature to the case statistics. From 18 UTC to 0 UTC (12pm to 6pm local) there is an increase in mean high cloud towards its maximum coverage. As sunset is approached at 0 UTC, high cloud area covers at least half the domain, with some cases exhibiting complete high cloud coverage.

As soon as the convective cloud begins to form at 18 UTC, the mean high cloud albedo rises to an average of ∼0.5, steadily increasing for the 7 hours thereafter. In this metric and from 18 UTC we see the greatest effect of the microphysics, with a
clear divergence in the means of at least two of the experiments. It is not until later in the day, at 0 UTC, that there starts to be an influence of the experiments on the high cloud area. And even so, this does not appear to be a strong or clear deviation.

The combination of the timing of solar irradiance, high cloud area and high cloud albedo determine the domain-average, high cloud SW CRE (Figure 5c). The peak occurs at 21 UTC (3pm local) with reasonable symmetry about that time. There are substantial CRE values, and variability, within the time period 18 to 0 UTC. This time period will be used for successive

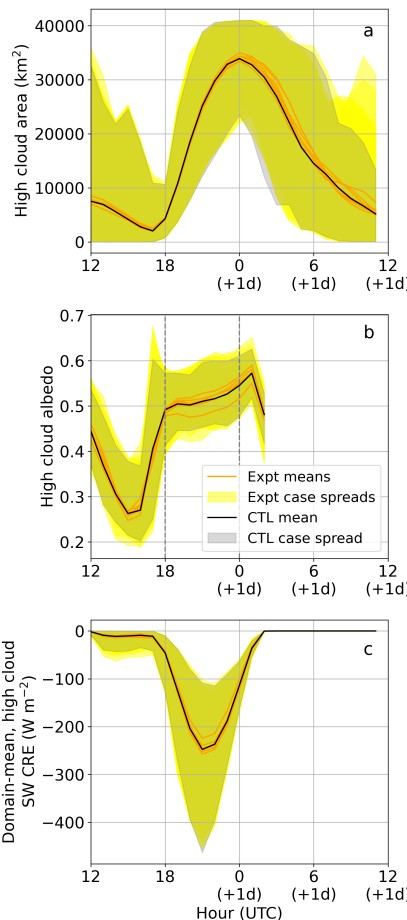

**Figure 5.** Simulated diurnal cycle of high cloud properties and radiation in 9 experiments, and 12 cases with similar timing of peak high cloud area (identified in Section 4, red circles on Figure 4c). From top to bottom: a) high cloud area, b) high cloud albedo, and c) domain average high-cloud SW CRE. There is a mean line for each experiment, and a shaded region around each line for the spread in case values for the experiment represented by the line. i.e. there's 9 lines, and 9 shaded regions surrounding them. Where the ctl and expts shading overlap, this appears as a murky yellow. "+1d" denotes the following day from when a simulation is initialised. High cloud albedo has no values after 2am because this is after sunset and solar irradiance is zero. For reference, when interpreting the high cloud area, the full analysis domain is ~41000 km$^2$.





analysis of albedo as it is a time period when albedo is able to influence the radiative effect, and also selects for the time period with clear differences in experiment mean high cloud albedo.

For the same three metrics as in Figure 5, now using daily averages for cloud area and high cloud SW CRE, and 18-0 UTC averages for albedo, Figure 6 presents the percentage anomaly for each experiment relative to the control. By using the percentage anomaly we account for the case-to-case variability driven by larger scale conditions discussed in Section 5.1.

Black dots represent each of the individual 12 cases, with the boxplots providing the statistics across the cases. The case spread in high cloud area makes it difficult to discern clear effects of the microphysics, but there are significant variations of the mean for the highest INP and droplet number cases, increasing mean high cloud by 6-8 % relative to the control. The large spread across cases of the effect on high cloud area (between -20 to +40%) suggests the cloud area variability is largely dominated by influences other than the perturbed factors in the sensitivity experiments.

For the high cloud albedo (Figure 6b), the spread is still apparent, but again there are significant effects found with some microphysical experiments. The significant effects on albedo arise from experiments varying cloud droplet number and INPs. For cloud droplet number, the influence on high cloud radiative properties manifests through its influence on homogeneously produced ice crystal number. The results show the response in this model is non-linear with absolute number. A reduction in droplet number of 125 x $10^6$ kg$^{-1}$ leads to a larger mean decrease in albedo compared to the increase in albedo seen with a

droplet number increase of 250 x $10^6$ kg$^{-1}$.

Both DCMEX INP (median and high) experiments increase albedo. The effects seen on high cloud SW CRE are broadly the same as with high cloud area and albedo, though drop500 and the INPmedian experiments don't meet the significance test for this metric, suggesting some degree of counteracting effects on area and albedo.

Other experiments show no significant effects in the various metrics for this set of cases and model. We will therefore not

study these in further detail. We do note that individual cases in other experiments can have large anomalies relative to the control. This acts as a warning regarding drawing conclusions from single case studies. However, it is also an encouragement to investigate such cases in more detail as long as they can be placed in the wider variability context.

## 5.3 Microphysical fingerprints

Figure 6 demonstrated that there are multiple potential microphysical causes of a anvil albedo change. If we were to see a

similar change in albedo in another model or observations, is there a simple approach to infer more about the cause of the albedo change?

Ice water path (IWP) can be used as a metric to distinguish between the droplet number and INP causes of albedo change. In particular, a threshold of 0.2 kg m$^{-2}$, which separates *thick* (high IWP) and *thin* (low IWP) cloud, as used by Sokol et al. (2024). Sokol et al. (2024) showed that this threshold aligns well with high cloud that has a neutral CRE. Meanwhile, thin

cloud has a net positive CRE and thick cloud has a net negative CRE. IWP is commonly available in model output and satellite products, and therefore offers a widely-applicable means for studying microphysical fingerprints.

Three metrics based on thick and thin high cloud allow us to decompose the anvil albedo change into a more detailed perspective:



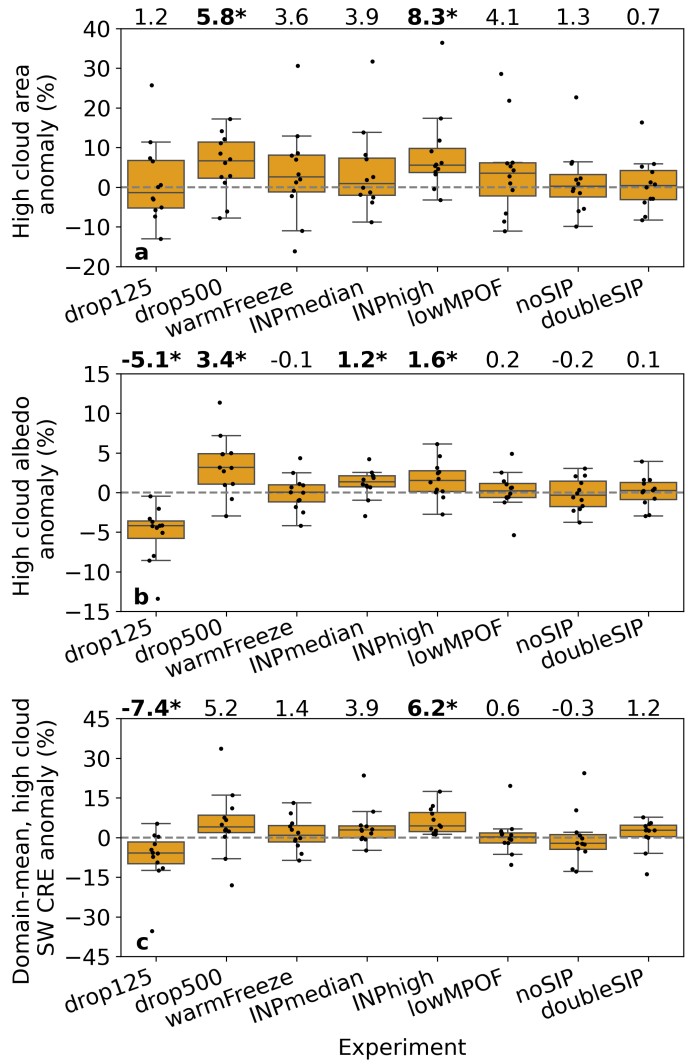

**Figure 6.** Percentage anomaly of high cloud properties and radiation of 12 cases for each of 8 perturbed microphysics experiments, relative to the control. Top panel shows 12–12 UTC mean high cloud area, middle panel shows 18–0 UTC mean high cloud albedo, and bottom row shows 12–12 UTC mean, domain-average, high-cloud, SW CRE. Boxplots show the median as a black horizontal mark, the upper quartiles as the edges of the wider box, and the whiskers show the minimum and maximum of the data points (excluding "outliers", defined as those more than 1.5 times the interquartile range from the nearest quartile on the box plot). Individual cases are black points. Numbers at the top of each panel give the mean percentage anomaly across cases of the experiment above which they sit. An asterisk and bold text marks those means that are significant using a 1-sample t-test and 5 % level. Mean and standard deviation of the control experiment absolute values across cases for each of panels a-c are $15000 \pm 5400$ km$^2$, $0.51 \pm 0.035$ and $-52 \pm 21$ W m$^{-2}$, respectively.



1. Thick-to-thin high cloud area ratio

2. Mean thick high cloud albedo

3. Mean thin high cloud albedo

Broadly speaking, the combination of the three metrics should account for changes in total high-cloud albedo change. Because thick cloud has a higher mean albedo than thin cloud, if metric 1 increases then the total high cloud albedo will correspondingly increase. Alternatively, if metric 1 stays constant but either metric 2 or 3 changes, then there will be corresponding change in

total high cloud albedo.

Figure 7 presents the distributions of high-cloud IWP and the three new metrics. Figure 7a shows that the IWP probability over the 12-case subset used in Section 5.2 is similar for the control and the 4 microphysical experiments that increase anvil albedo. However, there is indication that the INP experiments shift the distribution towards higher IWP (thick clouds), while the low cloud droplet number shifts the distribution towards lower IWP (thin clouds). Figure 7b provides the differences between

the experiment and control distributions, and confirms this interpretation, with higher occurrence of high IWP high cloud in experiments that increase total anvil albedo (i.e. drop500, INPmedian and INPhigh), and higher occurrence of thin cloud in the experiment that decreases total anvil albedo (i.e. drop125).

Figure 7c takes the same format as Figure 6 but shows the percentage anomaly of thick-to-thin high cloud area ratio. The mean changes in the experiments are consistent with the IWP distribution shifts. However, the changes are not significant

across the cases, which have a large spread in this metric. Consequently, this metric cannot be used to discern the differing microphysical causes of total anvil albedo change in our results, nor can it explain the significant microphysical influence on anvil albedo shown in Figure 6.

Figures 7d and e show the percentage anomaly in mean thick cloud albedo and mean thin cloud albedo. Together, the significant results from these two metrics can explain and discern the two microphysical causes of total anvil albedo change

in our results. All experiments influence the thick cloud albedo significantly (Figure 7d), and with the same sign as their total albedo change. However, only the droplet number experiments influence the thin high cloud mean albedo (Figure 7e), thereby providing a *fingerprint* that discerns this microphysical influence from the INP influence.

The metrics aggregate the radiative impact of changes in hydrometer populations. Figure 8 shows the mean in-cloud, combined ice and snow, specific number and mass profiles for thick and thin high cloud, for each of the control and 4 experiments

that significantly affect total anvil albedo.

The combination of ice crystal and snow specific number and mass changes determine the anvil cloud albedo changes. We note that *snow* is broadly defined in this 5-hydrometeor species model; it encompasses any ice habit that is not small ice crystals, nor rimed ice (which would be simulated as graupel). Graupel does not affect radiation in the model. The thick high cloud combined crystal and snow mass profile (Figure 8a) only has significant variation (marked with scatter points)

in the INP experiments, with the increase coming from a change in snow mass (Supplementary Figure S10). The combined number profile shows significant changes in all experiments (Figure 8c), consistent with how the experiments modify thick cloud albedo (Figure 7d). The droplet experiments influence specific ice crystal number throughout the full depth of high cloud



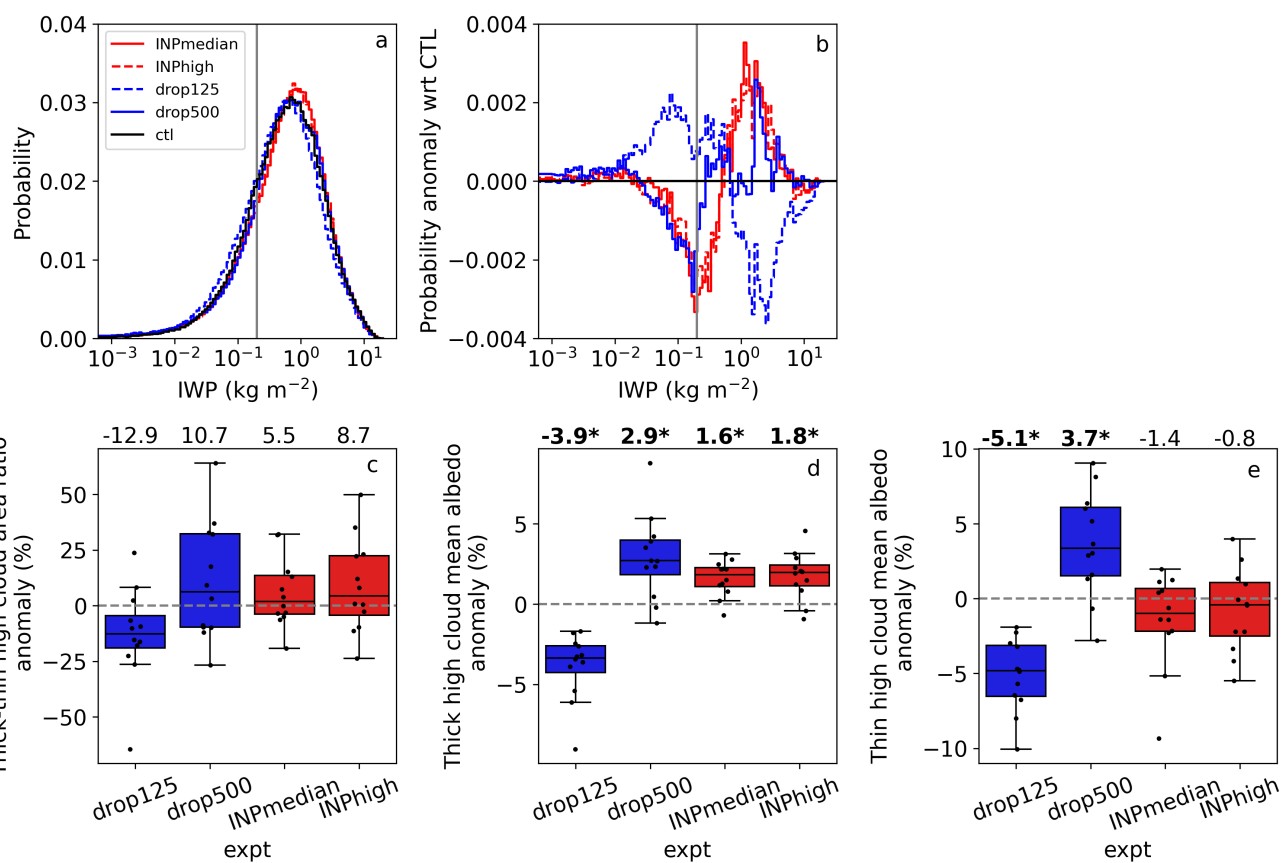

**Figure 7.** Ice water path (IWP) distributions and case-experiment variability of percentage anomaly of three *fingerprint* metrics. Panels c-e follow the same format as Figure 6. All metrics are calculated as averages over the 18-0 UTC period in order to relate to the effects on albedo presented in Figure 6b. Mean and standard deviation of the control experiment absolute values across cases for each of panels c-e are $3.2 \pm 1.6$, $0.56 \pm 0.03$ and $0.38 \pm 0.016$, respectively.



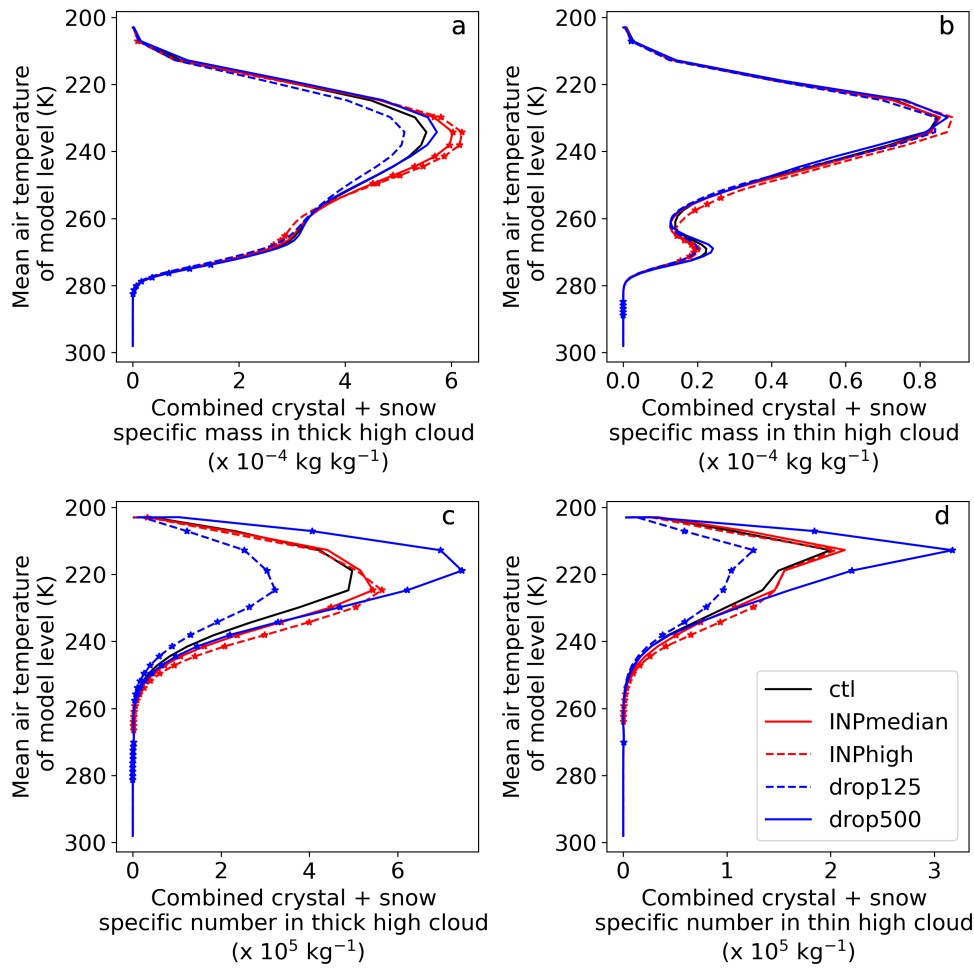

**Figure 8.** Combined ice crystal and snow mean in-cloud profiles for thick and thin cloud. Panels a and b are specific mass in thick and thin cloud, respectively. Panels c and d are specific number in thick and thin cloud, respectively. Scatter points denote where the mean experiment value is significantly different from the control at the 5% level using a 2-tailed, paired t-test. Equivalent figures for separate crystal and snow profiles are given in Supplementary Figures S9-10.





(Supplementary Figure S9), whilst the INP experiments influence snow number mainly through lower parts of the high cloud
(at temperatures > 220 K) (Supplementary Figure S10).

The thin cloud mass profile is largely unaffected in the microphysical experiments (Figure 8b). There are some significant changes low down in the profile of the INPmedian experiment but these are at temperatures indicative of mid-level cloud and therefore unlikely to be affecting anvil albedo. The decreased/increased droplet number experiments again cause decreases/increases in ice number throughout the profile (Figure 8d), consistent with decreased/increased thin cloud albedo in those experiments (Figure 7e). The INP experiments do increase number significantly at some levels, but these are not near the peak

in ice particle number at ∼210 K, and therefore, given no significant changes in thin cloud mass profiles, INP changes are unlikely to modify albedo substantially. This conclusion is corroborated by the insignificant influence of those experiments on thin cloud albedo (Figure 7e), and is consistent with a dominance of homogeneous freezing processes across the range of INP experiments explored here.

Figure 9 summarises the results of Figures 7-8 for the two main microphysical causes that increase total anvil albedo. In the

*Control* panel a number of illustrative components are shown. The orange arrows indicate the relative albedo of thick and thin cloud. The thick cloud arrow is made thicker because the albedo of thick cloud is higher than thin cloud (i.e. on average 0.56 vs 0.38). A number of nominal ice hydrometeors are shown in the high cloud (blue stars). The thick and thin parts of the high cloud are labelled.

The *high droplet number* experiment in Figure 9 does not exhibit significant changes to thick-to-thin high cloud ratio (Figure

7c) but increases crystal numbers throughout the high cloud (Figure 8c and d), leading to increased albedo (thicker orange arrows) throughout (Figure 7d and e), relative to the control experiment. With no significant reduction in combined crystal and snow mass content (Figure 8a and b) we infer the mass of each ice hydrometeor reduces in proportion to the increasing number of hydrometeors, and so reduce the ice hydrometeor size in the diagram.

The *high ice nucleating particle* experiment also has no significant change in thick-to-thin cloud ratio (Figure 7c). Since the

number and mass of snow hydrometeors increases in the thick cloud in this experiment, we add two larger ice hydrometeors to the diagram. This change corresponds to the increase in thick cloud albedo (Figure 7d). There is no significant change in thin cloud albedo so this part of the cloud is left unchanged.

## 6    Discussion

### 6.1    Microphysical controls on high cloud

Our work complements that of Fan et al. (2013), who carried out analysis of month-long simulations of convective cloud radiative properties over the southern Great Plains of the USA, including clean and polluted (i.e. high cloud condensation nuclei (CCN)) experiments. In addition, they considered a tropical western Pacific domain and a southeast China domain, which in many ways are demonstrated to respond similarly to southern Great Plains convection. Our domain relates to the southern Great Plains in that synoptic conditions affecting humidity variability are likely to be similar.





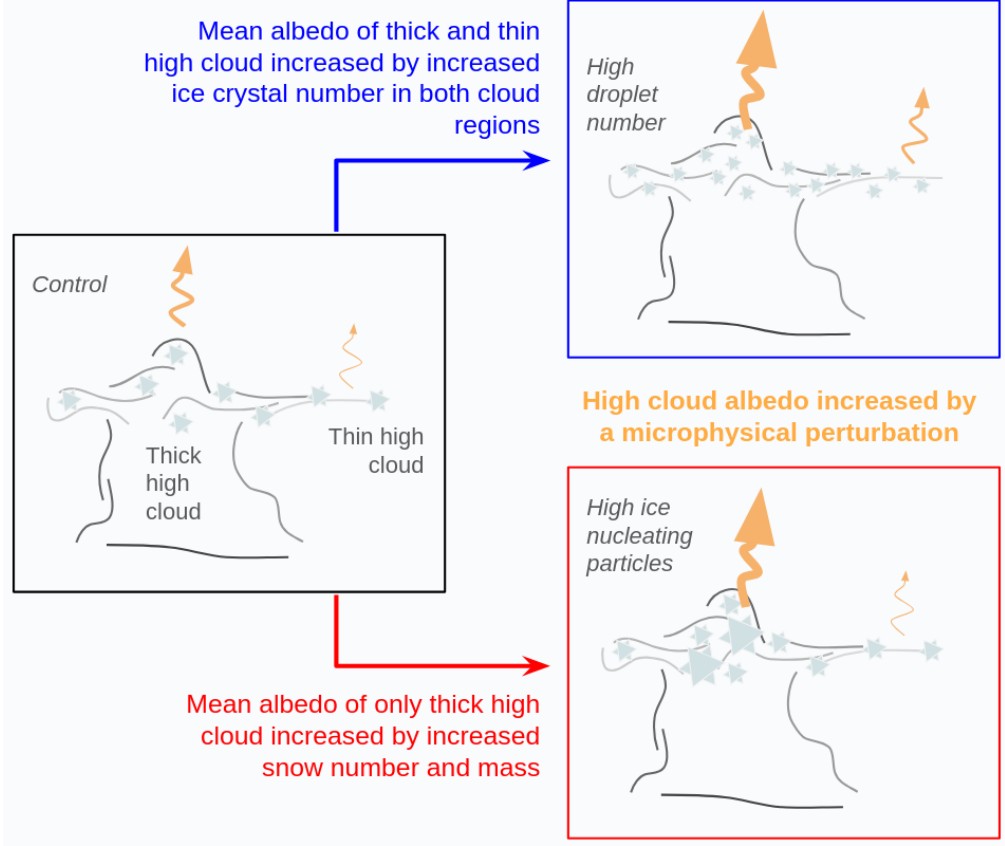

**Figure 9.** Diagram of microphysical causes of increased anvil cloud albedo in the UM-CASIM simulations. The diagram is qualitative; lengths, thicknesses, numbers and ratios are not to scale but representative of significant differences between the experiments as shown quantitatively in Figures 6-8. The thickness of orange wavy arrows represents mean albedo strength. Combined ice crystal and snow number are represented by the pale blue stars. The size of these symbols indicates the approximate mass/size of the hydrometeors.

Relevant results from our analysis are similar to that of Fan et al. (2013), though in some instances there are quantitative differences. We find that doubled cloud droplet number increases high cloud area by 6 % on average, whereas the six times increase in CCN of Fan et al. (2013) results in a 30 % increase in high cloud area (linearly equivalent to a 10% increase for doubled CCN). We do see a number of cases with increased cloud area by greater than or equal to 10 %, suggesting that this quantitative difference could result from different environmental conditions during the time periods used in each study. This is

feasible as both this study and Fan et al. (2013) find a dependence of cloud area on relative humidity, which will vary year to year.

     With respect to cloud radiative effect, Fan et al. (2013) found that a sixfold increase in CCN led to an increase in SW CRE across all three of their domains. The Great Plains simulations showed a 25 % smaller increase in SW CRE compared to the other two domains, suggesting that our region and results may underestimate the microphysical influence compared to what



might occur in other regions. Saleeby et al. (2016) also find evidence of higher aerosol concentration increasing high cloud area and albedo. Finally, further weight is added to the effect of cloud droplet number on convective cloud CRE as Miltenberger et al. (2018b), who used an ensemble approach for a single case of a sea breeze convergence line, found an increase of outgoing SW radiation with increasing droplet number.

Our work builds further on this past literature by exploring a variety of microphysical processes. Ice nucleating particles are

also found to produce a significant influence on anvil radiative effect in our results, but there are fewer examples of such studies in the literature, and none using a case-ensemble approach, as far as we are aware. Nevertheless, we consider here a selection of studies which do simulate INP effects on anvil cloud CRE for individual cases (Takeishi and Storelvmo, 2018; Hawker et al., 2021b, a). Hawker et al. (2021b) showed that anvil ice crystal number concentration can be influenced by the simulated temperature dependence of INP, with steeper gradients leading to less removal of water lower in the clouds and a greater number

of ice crystals in the anvil. Hawker et al. (2021a) went on to show that enhanced primary ice production in the upper parts of the mixed-phase region can remove liquid water, preventing homogeneous freezing so that the simulated anvil cloud properties are then determined by heterogeneous ice nucleation. In the present study homogeneous nucleation is always active, which means there is strong dependence of anvil cloud ice number concentration on cloud droplet number concentrations. Takeishi and Storelvmo (2018) looked at increasing INP and albedo response and find that albedo decreases with increasing INP.

However, this is only really apparent once they increase INP numbers by a factor of 100-1000, thereby making it competitive with the homogeneous freezing process which is heavily influenced by cloud droplet number. Similarly, Deng et al. (2018) found that homogeneous freezing is dominant in deep convective clouds until INP was increased by two or more orders of magnitude. For our environment, DCMEX observations show that the widely used Cooper (1986) parametrisation is a reasonable representation of the measured INP spectra, though the upper bound of measurements are more than ten times higher

than the Cooper curve (Figure 2). Therefore, our experiments focused on changes that increased INP by similar magnitudes and thereby demonstrate that INP increases could have significant impacts on anvil CRE within this uncertainty range. Nevertheless, having the measured INP concentrations to guide the choice of INP paramaterisations used in the modelling experiments removed a great deal of uncertainty in primary ice production. This allowed us to focus on the effect of natural variability in INP rather than the large uncertainty introduced if arbitrarily choosing an INP parameterisation from the literature.

The most comparable study to our own comes from Hawker et al. (2021b) and their *N12* simulation compared to *C86* simulation. The *N12* simulation has generally higher INP than *C86*, particularly at colder temperatures, and *C86* is based on the Cooper (1986) parametrisation used in our control simulation. Hawker et al. (2021b) uses a similar UM-CASIM configuration to this study and finds that the *N12* simulation increases domain outgoing shortwave as a result of increased total cloud fraction. However, the increased cloud fraction was not due to increased anvil cloud area, but increased lower-level cloud cover. It could

be important that high cloud does not appear to dominate the domain used by Hawker et al. (2021b), whereas it does in our simulations (Figure 5a). The Hawker et al. (2021b) study is also focused on a maritime, opposed to a continental environment. Some experiments in their study do increase the anvil cloud area or cloud albedo. However, the mixed responses of area and albedo may in part be due to use of a single case, as we find much case-variability across both high cloud area and albedo in the INP experiments of Figure 6. Or it may relate to the focus on a different convective cloud type (they study a mesoscale





convective system), or because their radiation scheme considers only hydrometeor mass not number. We have shown that snow specific number increases are relevant for INP affect on albedo in the UM-CASIM model. Hawker et al. (2021a) highlight that temperature dependence of INP can influence the anvil ice crystal size, and given our results suggest both an increase in snow number and mass, the temperature-dependent effects of INP on anvil radiative properties does warrant further investigation in future.

Hawker et al. (2021b) also perturbs the SIP (Hallett-Mossop only) rate and generally finds smaller effects than with the INP perturbed experiments, consistent with the lack of significance we find in our noSIP experiment (Figure 6). Overall, it is difficult to draw robust comparisons with single-case studies given the large case-variability that we have shown. In addition, SIP processes are still highly uncertain, with several potential processes proposed, each acting in different parts of the cloud and under different conditions (Field et al., 2017; Korolev and Leisner, 2020). The focus of Hawker et al. (2021a), Hawker

et al. (2021b) and this study has been on the Hallett-Mossop process, the most studied SIP process. However, since some SIP processes act at colder temperatures in the cloud than the Hallett-Mossop process (e.g. James et al., 2021; Zhao and Liu, 2021; Sotiropoulou et al., 2024), they may more readily influence the anvil hydrometeor composition. Future work should look to better observationally quantify SIP processes and establish if the details of SIP implementation are important for anvil radiative effect.

We have demonstrated how, in the model here, the influences of cloud droplet number and INP perturbations on high cloud albedo are distinct. This has been done using a simple IWP-based threshold to separately characterise thick and thin high cloud albedo. The three metrics we have presented can be calculated with commonly used model and satellite-based observations, and as such offer a means to further investigate high cloud radiative effects. There are a number of avenues that we propose for future exploration:

– To what extent are our *fingerprint* results reproduced in other models and regions?

– In observations, where it is more difficult to separately control for cloud droplet number and INP variation, does the cloud droplet influence mask the INP influence in thick high cloud?

– Do other variables, perhaps non-microphysical, have fingerprints discernable by the metrics proposed?

– What controls the thick to thin high cloud area ratio (given that no significant controls were identified in this study)?

## 6.2 Wider atmospheric associations with high cloud

Beyond the quantification of microphysical influence, our study has explored potential environmental conditions affecting high cloud radiative effects. This relates to studies of *cloud controlling factors* (CCF), which are large-scale environmental variables that influence CRE (Ceppi and Nowack, 2021). Our results highlight relative humidity and CAPE as potentially skilful cloud controlling factors (i.e. influencing cloud radiative properties) for this cloud type (Table 2). Meanwhile we found that wind

shear, and even the simpler wind speed at 200hPa, did not add value for the metrics studied in observations, but did add value for predicting modelled SW CRE.



Both Andersen et al. (2023) and Wilson Kemsley et al. (2024) use relative humidity at 700hPa (i.e. R700) in their regressions as this is already an established factor. Our work here, therefore, adds impetus to the value of that metric for high cloud SW CRE and albedo. Wilson Kemsley et al. (2024) explores the use of CAPE but does not find it is the most skilful of metrics when predicting high cloud LW CRE. However, we found that CAPE was better able to predict albedo (a feature that affects SW CRE) than anvil area (a feature that affects LW as well as SW). However, R700 was the best predictor for SW CRE, which is the culmination of albedo and area variability. In particular, the observation-based regression found CAPE to be one of the most valuable predictors of anvil albedo. Since Andersen et al. (2023) did not use CAPE, we believe this factor has so far gone overlooked with regard to high cloud SW CRE, and therefore encourage its testing in future CCF work.

The link between wind shear, anvil cloud area and optical depth has a history of study in the literature (e.g. Erickson, 1964; Lin and Mapes, 2004). Andersen et al. (2023) do seem to have tested 300 hPa winds as a cloud controlling factor, but have not presented sensitivities for them, so we assume they did not find these factors valuable. Wilson Kemsley et al. (2024) did find 300 hPa vertical wind shear to be a useful metric for high cloud LW CRE. Our observation-based results do not highlight upper level wind speeds (V200) as significantly impacting anvil cloud radiation, but we may be on the threshold of sufficient sample size to explore this as the modelling results do suggest upper level wind speed to be a useful secondary predictor of SW CRE, especially when combined with R700. More investigation of the role of dynamics on anvil cloud radiative effect is warranted.

The more that investigations of anvil cloud feedback come to focus on SW CRE and albedo, the more important it is to recognise that the timing and life-cycle of clouds, in relation to the solar cycle, is integral. In this study, we have endeavoured to control for such diurnal variability through selecting a subset of cases which have similar timing of maximum high cloud. This has proven a useful approach for the New Mexico domain because the convective storms are tightly timed with their trigger of the elevated heating, which itself is relatively consistent day-to-day. However, in regions with longer-lived storms, or where convective triggers vary more in timing, a more complex approach may be needed. Gasparini et al. (2021) and Jones et al. (2024) have demonstrated the value in a Lagrangian methodology for studying anvil CRE. Whilst Jones et al. (2024) found a net CRE close to zero for their observed cases, a shift in the timing of convection with global warming could introduce a currently un-characterised anvil feedback, where land convection may prove to play a more important role (Liu et al., 2025).

The DCMEX project is part of a UK Natural Environment Research Council strategic programme, CloudSense. A number of studies from that programme provide new information regarding circulation interaction with high cloud feedbacks (Mackie and Byrne, 2023; Hill et al., 2023; Natchiar et al., 2024). In particular, Natchiar et al. (2024) uses aquaplanet simulations to describe how microphysical processes interact with circulation changes to control tropical high cloud area. Our results add further to this in highlighting the importance of certain environmental conditions and microphysical processes in accurately simulating anvil cloud albedo and CRE.

# 7 Conclusions

This study has drawn on observations from the DCMEX field campaign to investigate controls on anvil cloud albedo with a double-moment cloud microphysics scheme in a convection-permitting model. The focus is on a region of New Mexico, USA,





where orographic, thermal triggering of deep convection reliably occurs during summer months. The convection-permitting model used is able to capture the variability in cloud morphology and radiation across the campaign period.

Our results describe two distinct microphysical perturbations which significantly impact anvil cloud albedo and radiative effect in the model. Namely, through increased cloud droplet number or INPs. These experiments, which spanned the observed droplet (2x) and INP ($\sim$10x) variability in DCMEX aircraft measurements, showed average increases in high cloud albedo of

approximately 1-3 %. This level of change in mean albedo over the high cloud area can result in significant effects on cloud radiative effect, and therefore our results encourage continued development of accurate droplet life-cycle and INP representation within atmospheric models in order to accurately simulate anvil cloud influence on radiation budgets.

An IWP threshold is used to decompose the anvil albedo change into three metrics that together provide *fingerprints* of the identified microphysical causes of albedo change, i.e. these metrics allow the cause to be discerned. For the experiment with

increased droplet number, the mean albedo of both *thick* (high IWP) and *thin* (low IWP) high cloud is increased. But for the increased INP experiment, only the thick cloud albedo is significantly affected.

The hydrometeor profiles of mass and number show that increasing droplet number increases the ice crystal number throughout thick and thin high cloud. This highlights the dominance of homogeneous freezing in determining anvil cloud ice number. However, increasing the heterogeneous freezing rates through higher INP can affect the thick portion of the high cloud. It does

this in the simulations here by increasing both the number and mass of snow hydrometeors.

As well as quantifying direct microphysical influences on anvil albedo, we have also presented associations between large-scale environmental factors and anvil properties. Anvil cloud albedo, area and SW CRE are all significantly correlated with relative humidity at 700 hPa across the campaign cases. The observations additionally suggest that CAPE may influence anvil albedo, while the model suggests that upper level winds could play a secondary role in the variation of SW CRE itself.

In summary, this study has described a new approach for exploring variability in high cloud albedo in the form of fingerprint metrics. It has also provided evidence, on the scale of cloud systems, of large-scale environmental factors that may influence radiative properties of anvil cloud. While the application here of the fingerprint metrics has been to experiments perturbing model physics, the approach may also prove useful if compositing cloud from different environments. For instance, if studying satellite-observed anvil cloud from low and high aerosol environments. Meanwhile, the anvil cloud property with highest

correlation with environmental factors was the anvil albedo. This encourages investigation of the potential for cloud controlling factor analysis, or theoretical approaches based on large-scale features, to predict the anvil albedo feedback, which is currently the least well understood aspect of anvil cloud feedback.

*Data availability.* DCMEX campaign data, including aircraft data, are available at
https://catalogue.ceda.ac.uk/uuid/b1211ad185e24b488d41dd98f957506c (Facility for Airborne Atmospheric Measurements et al., 2022).

INP filter data are available at https://doi.org/10.5518/1476 (Daily et al., 2024). ERA5 data were accessed through the Climate Data Store (Accessed: 7 Feburary 2025): https://doi.org/10.24381/cds.adbb2d47 (single level data) and https://doi.org/10.24381/cds.bd0915c6 (pressure level data). CERES data was downloaded from the NASA LaRC webpage https://ceres.larc.nasa.gov/data/ (Accessed: 7 February



2025). GOES data were downloaded using the goes2go python package (Blaylock, 2023). Model data was archived on the UK Met Office MASS archive under suite ID u-dd455, and a portion of the data encompassing that used in this study is available at

https://catalogue.ceda.ac.uk/uuid/b1211ad185e24b488d41dd98f957506c (Finney, 2025).

*Author contributions.* DLF led the data generation, analysis and writing of the study. AMB led the successful funding application, conceptualised the study, and supported interpretation of results and writing. PRF supported conceptualisation, data generation with the UM-CASIM, interpretation and reviewing of drafts. MID generated the INP parametrisation for application in the model. BJM supported INP parametrisation development and supported interpretation of results and reviewing of drafts. MS, PJC, ZC and SB supported with interpretation of
results and reviewing of drafts.

*Competing interests.* The authors declare that they have no conflict of interest.

*Acknowledgements.* This work was supported by the Natural Environmental Research Council (NERC; NE/T006420/1, NE/T006439/1 and NE/T00648X/1). We are grateful to all at FAAM, Airtask, and Avalon that made the DCMEX flying campaign possible. Airborne data were obtained using the BAe-146-301 Atmospheric Research Aircraft [ARA] flown by Airtask Ltd and managed by FAAM Airborne Laboratory.
The FAAM Airborne Laboratory's research aircraft is owned by UK Research and Innovation and the Natural Environmental Research Council. It is managed through the National Centre for Atmospheric Science, and leased through the University of Leeds. Analysis for this study was carried on onJASMIN, the UK's collaborative data analysis environment https://www.jasmin.ac.uk. The authors would like to thank Dr Steve Abel and Dr Matt Evans for their provision of the observation-based mixed-phase overlap fraction parameter estimate, and their advice on the manuscript.



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



**Table 1.** Model experiment overview. The feature of each experiment that differs from the control is shown in bold. Abbreviation *Het* refers to *Heterogeneous.*

| Experiment short name | Description | Cloud droplet number (× 10⁶ kg−1) | Het. freezing temperature (°C) | Primary ice scheme | mixed-phase overlap fraction | SIP rate (× 10⁶ kg−1) |
|---|---|---|---|---|---|---|
| **ctl** | control setup | 250 | -8 | Cooper | 0.85 | 350 |
| **drop125** | lowest cloud droplet number | **125** | -8 | Cooper | 0.85 | 350 |
| **drop500** | highest cloud droplet number | **500** | -8 | Cooper | 0.85 | 350 |
| **warmFreeze** | warmer ice nucleation | 250 | **-5** | Cooper | 0.85 | 350 |
| **INPmedian** | median DCMEX INP curve | 250 | -8 | **DCMEX median** | 0.85 | 350 |
| **INPhigh** | high DCMEX INP curve | 250 | -8 | **DCMEX high** | 0.85 | 350 |
| **lowMPOF** | lower overlap fraction | 250 | -8 | Cooper | **0.5** | 350 |
| **noSIP** | no SIP | 250 | -8 | Cooper | 0.85 | **0** |
| **doubleSIP** | 2 x SIP rate | 250 | -8 | Cooper | 0.85 | **700** |





**Table 2.** Summary of model and CERES-FBCT (obs) adjusted $R^2$ from linear and multi-linear regression fits of the feature(s) against three high-cloud metrics (columns). Feature values used in the regressions were daily means. Target values were 16-22 UTC means for albedo and cloud area, and were approximate, daily, domain mean, for high cloud, SW CRE (see Section 3 for details). Asterisks denote adjusted $R^2$ values with a p-value<0.05. Whereas the table numbers are the adjusted $R^2$, colours separately denote the sign of the regression coefficient for single feature variable regressions that are significant, with red and blue representing positive and negative, respectively. We note that adjusted $R^2$ can be negative if the conventional $R^2$ is close to zero, which is the case for some elements of the table.

| Feature(s) | Albedo | | Cloud area | | Approx. daily SW CRE | |
|---|---|---|---|---|---|---|
| | model | obs | model | obs | model | obs |
| R700 | 0.44 * | 0.59 * | 0.13 | 0.38 * | 0.23 * | 0.53 * |
| CAPE | 0.19 * | 0.68 * | 0.02 | 0.22 * | 0.01 | 0.38 * |
| V200 | -0.05 | 0.03 | 0.03 | 0.00 | 0.16 * | -0.01 |
| R700, CAPE | 0.41 * | 0.70 * | 0.09 | 0.36 * | 0.24 * | 0.52 * |
| R700, V200 | 0.41 * | 0.60 * | 0.19 | 0.37 * | 0.40 * | 0.53 * |
| R700, CAPE, V200 | 0.38 * | 0.70 * | 0.14 | 0.35 * | 0.40 * | 0.51 * |