# Peer review of "Microphysical fingerprints in anvil cloud albedo"

_EGUsphere, 2025_

## Author Comment (AC1)

*Thank you to the editor and reviewers for their time. We respond below in green italics to comments. Line numbers refer to line numbers of the marked-up manuscript.*

*We have identified corrections in addition to the reviewer comments. Some typos and incorrect units and formula were identified in the INP parametrisation description. These were only errors in the text - simulations had been run with the correct method, as now described in the new manuscript submission. Corrections to the following have been made:*
1. *Parameter "a" should not have a minus sign (L172 And L173)*
2. *Equation 1 RHS first term should be $n_s$ not $10^{ns}$ as per rearrangement of fig 7 caption of Harrison et al. (2019) https://acp.copernicus.org/articles/19/11343/2019/*
3. *$A_{dust}$ should be units of $cm^2\ L^{-1}$ in order to have consistent units across the terms. The conversion has been made to the text (L168 and L174)*
4. *An aerosol condition of 200 $cm^{-3}$ was used (fig2 caption)*

*We have also made a technical correction of "ice nucleating particle" to "ice-nucleating particle" throughout.*

Blaž Gasparini
Finney et al. use observation-informed cloud-resolving modeling to investigate how both large-scale environmental conditions and microphysical properties influence anvil cloud albedo. Their cloud-resolving model simulations, constrained by observations from the DCMEX campaign of orographically forced convection in New Mexico, USA, reproduce the observed cloud and radiative properties in a reasonable way. The study reveals a substantial sensitivity of anvil cloud albedo to cloud droplet number concentration and, to a lesser degree, to ice-nucleating particle concentration.

The manuscript is clearly written, logically structured; it was easy to follow the line of thought and understand the key outcomes. However, I have several questions and suggestions that I believe should be addressed prior to publication. In particular, I encourage the authors to further explore the mechanisms underlying the reported changes in cloud properties.

*Thank you Dr Gasparini for your review. We respond to your questions and suggestions below.*

General comments

1.) Mechanisms and physical interpretation

While the impacts of CCN and INP perturbations on anvil clouds are clearly described, the manuscript would benefit from further elaboration on the underlying mechanism leading to these changes. How exactly do changes in CCN or INP influence cloud albedo? How do the CCN propagate to changes in ice phase clouds? Are the effects primarily driven by direct microphysical modifications (e.g., changes in ice crystal size or number), or are they mediated indirectly through changes in updraft dynamics? How do specific microphysical process rates respond to the perturbations? Or, if we take a step back: What are the dominant processes that determine ice crystal number and mass in these simulations/this types of high clouds? How do they compare to those in more frequent type of anvil clouds, e.g. tropical anvils?

*Following the reviewer's questions we have explored the potential indirect dynamical effect of the experiment perturbations on clouds through influencing latent heat release and therefore updraught velocities. We have studied a number of updraught metrics (mean, percentiles and maximum) for thick and thin high cloud profiles equivalent to Figure 8. We find no significant coherent effects in these profiles which therefore suggests no major role for an indirect dynamical effect. Therefore, we infer the influence of microphysical perturbations is dominantly a direct one, through influencing the ice particle size and concentration. Notably, increasing cloud droplet number increases the homogeneous ice crystal formation, while increasing INP increases frozen mass and number in the mixed phase region and lower to mid anvil. Since perturbing cloud droplet number directly affects the high cloud albedo, we infer that homogeneous freezing plays a major role in these clouds.*

*We have added the following text:*
*"Equivalent profiles of mean, 95th and 99th percentile updraught velocity were analysed but found to show no significant response to the microphysical perturbations. We therefore infer that the microphysical effects on high cloud albedo shown here are primarily through direct influences on the ice particle number and size, and not through indirect dynamical influence on latent heat release and buoyancy." (L516)*

2.) Robustness of results

Are the key results robust? How sensitive are the findings to stochastic variability? What would happen if one were to run e.g. an ensemble of 5-10 simulations with perturbed CCN and INP conditions?

*The focus of the DCMEX campaign approach of observing multiple storms under similar conditions, and this modelling setup of applying the same microphysical perturbations across multiple cases, is all driven by the aim to establish a robust methodology. Significant results presented here arise from robust mean responses across 12 cases. These 12 cases are 12 real cases, driven by similar convective triggering in the form of orographic heating. By studying variability across real cases we explore a wider range of variability than a meteorological ensemble for a single case, we therefore consider our approach more robust, and more representative of real-world variability, and a stronger test of statistical significance.*

*We have added the following sentence in order to draw this point out in the manuscript:*
*"Simulating 24 real-world cases, and focusing on 12 with similar convective forcing and timing, has allowed for a rigorous test of significance." (L661)*

3.) Connection to observations

The manuscript would benefit from a stronger connection between the modeling results and observed high cloud albedo changes. Is there satellite evidence of similar anvil albedo changes over the Magdalena mountains under comparable dynamical but different aerosol

conditions? Although the DCMEX campaign may not cover a long enough period to address this conclusively, long-term satellite records might offer additional context.

*In Figure 4e we had presented information regarding high cloud albedo variability in our domain during the DCMEX campaign. Based on the reviewer's suggestion, we explored combining this with section 5.1 results which found associations between albedo and environmental factors. By subtracting the observed relationships (notably the multilinear regression of CAPE and RH700 with albedo) from the observed high cloud albedo we can obtain a residual high cloud albedo variability. This residual should account for variability in albedo that results from the environmental factors tested which, according to Table 2, explains approximately 70% of the variability across all the cases. We can then explore whether a significant proportion of the residual variability can be explained by variability in CDNC and INP, as measured during the campaign.*

*It is not appropriate to use the whole time series. We apply the following conditions to ensure an appropriate subset of cases are used:*
1. *High cloud area must be greater than 5% of the domain to ensure there is a robust sample of cloud from which average high cloud albedo can be estimated.*
2. *The DCMEX aircraft must have flown, and sampled deep convective cloud.*
3. *At least 3 suitable INP filter measurements at either -15C or -25C from either the teflon or polycarbonate filters.*

*We choose -15C and -25C as two key marker temperatures of the INP temperature dependence shown in Figure 2a. Because INP concentrations are orders of magnitude different at these two temperatures, we standardise the concentrations relative to the variability within the measurements across cases for the given temperature and filter type. We take the mean of these standardised values to explore the variability as a predictor of high cloud albedo residual variability. We use the mean CDNC as the metric to represent variability in CDNC. Other metrics such as various percentiles were explored but these generally varied coherently with the mean. The std error spread (dotted red lines below) shows that the mean is reasonably well constrained by observations.*

[Figure]

**Figure**. Albedo variability against CDNC and INP variability. (Top) Residual albedo calculated as fig4e black line subtracting the effects of RH700 and CAPE using coefficients from the multi-linear regression for which adjusted $R^2$ was presented in Table 2. The panel is using CERES-FBCT albedo and ERA5 RH700 and CAPE data. (Middle) Mean, standard deviation and standard error of CDNC as calculated from 1Hz means measured by the CDP instrument on the DCMEX aircraft (data are only included for cases where sufficient INP data is also available for this comparison). (Bottom) INP from teflon and polycarbonate filters at -15C and -25C, standardised using the variability for the given temperature and filter type, with the black line showing the mean of the standardised points.

*To the eye, it looks as though the CDNC and INP variability may explain the residual albedo variability. However, when a multi-linear regression is fit, neither CDNC nor INP are significant predictors at the 5% level, and this is without accounting for reduced effective sample size resulting from temporal auto-correlation. It was explored whether relaxing the INP condition, and only regressing CDNC was able to generate a significant result. However, the 14 data points then available still did not provide a significant result.*

*We conclude that the 9-14 data points available here are not sufficient to achieve significant results for the observed microphysical perturbation effect on albedo. Otherwise, a more sophisticated method, beyond the scope of this study, is required to extract the relationships. Nevertheless, the time series of points is encouraging that an observed relationship may exist. A remote sensing, tethered balloon or drone-based observation technique of boundary layer and cloud based droplet number may generate sufficient sample size to compare to satellite high cloud albedo.*

*We have included the following text:*
*"It was also explored whether an observational-based analysis could robustly quantify the effect of CDNC and INP variability on high cloud albedo during the DCMEX campaign. This was done by calculating a residual high cloud albedo by subtracting the R700 and CAPE regression of Section 5.1 from the CERES high cloud albedo of Figure 4e. The residual high cloud albedo was regressed against mean CDNC and mean standardised INP at -15C and -25C measured with the DCMEX campaign aircraft across 9 suitable cases. A significant effect for INP and CDNC was not found, so we conclude that a larger sample size is likely needed to extend this work observationally." (L520-525)*

4.) Broader relevance

Although it may go beyond the scope of this study, the potential for generalizing these results is worth considering. Could e.g. long-term satellite retrievals combined with reanalysis data help assess the broader applicability of the findings? Additionally, is there an analogy between orographically driven convection and island-driven convection in the tropical Warm Pool?

*The existing discussion section implied that these results should be explored further in observations, but we have added a remark to explicitly propose future analysis with satellite data:*
*"Can satellite retrievals of aerosol be used to establish whether there is significant variability in high cloud albedo fingerprint metrics between high and low aerosol regions and times" (L615)*

*We can see your point regarding similarities in orographic vs land-ocean driven convection. Ultimately, we don't consider the triggering mechanism a constraining factor with regard to the microphysical effects determined here. It is primarily a means by which the DCMEX observational campaign was able to get multiple comparable cases with similar forcing. It doesn't feel like we can add any insight regarding the similarity to island convection beyond the well-understood knowledge around temperature and humidity gradients driving convection in both regions. Therefore, we have not added any text on this point.*

5.) Selection of meteorological predictors

The choice of meteorological variables and cloud-controlling factors used in the analysis is not entirely clear to me. Why did e.g. the authors exclude some of the cloud controlling factors that are thought to be useful in explaining high clouds at climatological timescales, e.g. the mid-tropospheric updraft, upper tropospheric stability?

*We expect mid-tropospheric updraught velocity to be broadly captured by our CAPE predictor. However, we agree with the reviewer that upper tropospheric stability would add a predictor not wholly represented by the existing set. It is a variable that has been associated with high cloud area and LW CRE in the literature. We have added it into table 2 along with text in the methods and results sections to introduce the variable and describe the results:*

*"Temperature profiles are used to calculate upper tropospheric static stability following the equation given by Wilson Kemsley et al. (2024). Through studying the temperature profiles across our cases, we find the tropopause occurs consistently around 100-125 hPa. Therefore, we take a fixed pressure level range, of 300 to 100 hPa, over which to calculate the mean upper tropospheric static stability." (L216-219)*

*and*

*"Whilst there is no significant fit to SUT on its own, the variable does provide marginal improvements in skill when combined with R700. The minor role of static stability in these results compared to the literature (Wilson Kemsley et al., 2024), may be due to the use of*

*daily temporal frequency here, compared to the widely studied monthly temporal frequency in the literature. However, we do find that some significant results become apparent with the SUT explanatory variable if regressions are calculated on data conditional on at least 5% high cloud being present (Supplementary Table S2). This further suggests the importance of other environmental conditions being in place in order for high cloud properties to become sensitive to SUT on daily temporal scales." (L384-390)*

*In adding additional rows, the table has become busier. To help the reader, we have added bold to cells with the highest value adjusted R2. Where there are multiple cells with the highest value, we bold the one with the fewest explanatory variables.*

*We note that in updating the table, we realised a slightly old version of the table had been included and so have updated some existing cell values too – these adjustments are small and are inconsequential to the conclusions.*

*Overall, upper tropospheric stability on its own does not provide a valuable prediction for any of our target variables, even high cloud area. However, when combined with R700 and more predictors it provides some cases of marginal improvement in adjusted R2.*

*Through our further consideration of this section of results, we have considered whether a regression based on cases conditional on more than 5% high cloud cover, instead of using the whole time series, could be insightful. Therefore, in a new supplementary table we present these regression results. These results are broadly consistent with the unconditional regressions but some target variables show a significant association with upper tropospheric stability on its own, so it seems valuable to include. This is noted in the new text shown above.*

6.) Longwave cloud radiative effect and related quantities (e.g. cloud top temperature)

Although the study focuses on shortwave albedo effects, additional discussion of longwave fluxes and related quantities such as cloud top temperature would be useful in bringin a more holistic view on anvil changes. For example Fig. 8 suggests possible changes in cloud top temperature. Moreover, given that cloud LW emissivity saturates at relatively low cloud optical depths (~2-3), LW fluxes are respond primarily to changes in thin anvils. Can the authors provide more insight on this aspect?

*To provide a more complete picture of the radiative effects we have added a supplementary figure, equivalent to Figure 6, but showing the effects of the experiments on cloud top temperature, cloud-average LW high cloud CRE, domain-average LW high cloud CRE and domain-average net high cloud CRE.*

*This shows that only the cloud droplet experiments significantly modify the cloud top temperature and cloud-average LW CRE. However, the significant domain-average LW CRE responses are dominated by the experiments with significant changes in cloud area (Figure 6a). This suggests that the cloud top temperature changes are not of a great enough magnitude to have a major role in the overall radiative response. The net CRE is a result of many spatially and temporally varying factors combining, and as such significant results in this metric probably require larger sample sizes. However, the reduced cloud droplet number*

*experiment does show a significant weakening of net CRE as a result of weakened domain-average SW CRE (Figure 6c), which resulted from reduced high cloud albedo (Figure 6a).*

*Given no significant changes in vertical velocity profiles, as commented on in another response, it is not clear what is driving the cooling of the cloud top temperature. We checked for significant changes in the daily mean profile of static stability profile, but this did not provide any insight. It may be related to changes in particle size distributions and therefore sedimentation rates – the average particle mass decreases with increased cloud droplet experiment. However, we have not looked to confirm this in any greater detail, as the CTT changes seem fairly inconsequential and it is not the focus of this paper.*

*We have added the following text:*
*"Cloud droplet experiments also have a small significant effect on the cloud top temperature, with increased cloud droplet number leading to cooler high cloud top temperature (Supplementary Figure S9). The effect on domain average high cloud LW CRE is insignificant, which is instead dominated by high cloud area changes (Figure 6a). As such we have not investigated this change in cloud top temperature in further detail, and instead focus on the albedo." (L439-443)*

[Figure]

Specific comments

1.) 70 vertical layers are rather few for correctly representing thin anvils responses to any kind of forcing. Would the results hold with higher vertical resolution in the upper troposphere? Testing or at least discussing this would add credibility to the conclusions.

*We used the standard operational setup that was available at the time so results here are traceable to operational evaluation. This setup has performed well in our evaluation. We have added Supplementary Table S1 so that readers can better understand our model vertical level setup in relation to this point.*

2.) The mechanisms by which CCN and INP perturbations affect anvil albedo appear relatively straightforward. Would similar sensitivities be found using a simpler, single-moment microphysics scheme? This question is especially relevant given that many global storm-resolving models use simplified microphysics. If such interactions are as robust as they appear, this would suggest that even models with basic microphysical representations might capture the essential response to aerosol perturbations. A comment on this would be helpful.

*In a single-moment scheme the size of particles is mostly dependent on the mass, so unless the mass changes substantially, then there wouldn't be an effect.*

*In our droplet experiments, for example, the profile of ice mass does not change significantly, only the number profile changes significantly. Therefore, this microphysical effect requires a two-moment scheme to simulate it.*

*We have added the following sentence as a comment on this:*
*"Through only significantly affecting the ice number profile, not mass profile, the droplet experiments highlight the value of using a two-moment microphysics scheme."(L498)*

Section 4: How are cases with multiple cloud layers handled in the analysis? For instance, what if two high cloud layers are present? Does this occur frequently, and if so, how is it treated in the retrievals or model evaluation?

*Neither analysis of the CERES data nor the UM-CASIM data has separated out lower-level cloud layers. It is assumed that if high cloud is present then it is the dominant effect on radiation. It is accepted that lower-level cloud layers will influence the cloud radiative effect to some degree. However, since the majority of the high cloud in these cases are "thick" with regard to IWP (Figure 7a), we consider this to be a secondary consideration.*

*We have added text to the methods section to make sure this is clear:*
*"High cloud in model columns and in CERES satellite retrievals are defined by the occurrence of high cloud, with the presence or not of lower level cloud layers not being considered. This means that the radiative properties of high cloud will encompass multi-layer interactions where such layers are present." (L238-240)*

Data availability: I think the links don't have the model data uploaded, if I understand the website contents correctly.

*Apologies, the wrong link was inserted. It should have been*
*https://catalogue.ceda.ac.uk/uuid/b850297a4de4493b8ff048f574811e25/*

Best regards,

Blaž Gasparini

Anonymous Referee 2
Review of manuscript: "Microphysical fingerprints in anvil cloud albedo" by Finney et al.

General comment:

This manuscript addresses environmental and microphysical effects on SW and LW cloud radiative forcing via changes to cloud thickness and area via a modeling study of topographically forced convection over the southwest United States. Simulations with the UM model appear to reasonably represent the modes of convection and associated cloud cover. Sensitivity experiments suggest that variations in droplet number and INP have the greatest impact amongst their sensitivity tests. The paper is well written, and the explanations are easy to follow. There are, however, a number of comments below that should be addressed related to the model capabilities and the assessment of microphysical responses to changes in droplet number and INPs.

*Thank you for your review. We respond to your comments below.*

Specific comments:

1.Lines 99 and 105: How can you use a 75 second timestep with domain grid cell spacing of 1.5km and not encounter CFL errors, especially in the vertical? Also, given the rapid changes that can occur in clouds and their impact on radiation, a 15-minute time-step for radiation updates seems very long.

*We have added the following text which explains how the 75 second timestep is feasible:*
*"The relatively long model time-step is feasible due to the following approaches used in the dynamical and microphysical components of the model. The model dynamics is semi-Lagrangian, semi-implicit (Wood et al. 2014) that allows for longer time-steps than Eulerian advection for atmospheric prognostics. For sedimentation of hydrometeors we make use of the approach of Rotstayn (1997) and outlined in A.12 of Field et al. 2023. The method applies an exponential filter, to maintain stability, that increases in strength as Courant number increases. In practice the filter only becomes important for the lowest levels in the model." (L109-114)*

*Thank you for highlighting the radiation step, we should add more detail on this. The full radiation scheme is run every 15 minutes, but cloud radiation effect is calculated on a 5 minute timestep. We have added this point to the model description (L106)*

2.Lines 125-140: Are the droplet numbers constant over time? Or do the droplets undergo autoconversion, accretion, riming, homogeneous freezing, etc? This is rather critical since vertical transport of droplets to the anvil level and subsequent homogeneous freezing to generate high concentrations of small anvil ice can have a substantial impact on cloud top albedo.

*Our experiment design has used a fixed vertical profile of cloud droplet number (Figure 2a, red and orange lines). This decreases with height broadly following the aircraft observed measurements (fig 2a), and then decreases exponentially, to approximate processes which reduce number. If liquid cloud fraction is less than one then the grid cell mean cloud droplet number decreases proportionally. We choose this approach to ensure clear attributability to the cloud droplet changes, opposed to more complex interactions with aerosols if these were applied interactively.*

*We have added additional text to the methods section to ensure all the points above are made, and that our rationale is clear:*
*"Cloud droplet number concentration is largely determined by aerosol emission and transport. However, to constrain our experiments to focus on the in-cloud processes, and ensure changes are attributable to the cloud droplet perturbations, we prescribe a fixed profile of cloud droplet number concentration. This is defined by a constant droplet number per kilogram of air, a height above ground level at which droplet number exponentially decays, and the exponential decay rate. Grid cell mean cloud droplet number scales proportionally with liquid cloud fraction of the grid cell." (L133-136)*

3.Lines 219-220: Did you test cloud mass mixing ratio thresholds other than the one stated here? This threshold is rather low and may not constitute a visually apparent cloud. Does the satellite imagery use a similar sort of threshold for determining cloud presence?

*Following the reviewer's question we have tested higher thresholds of $5x10^{-6}$ and $1x10^{-5}$ kg/kg. However, for a set of sample cases, this had no effect. The reason is that our condition requiring a cloud fraction of 1 dominates, and would need to be relaxed in order for cloud mass thresholds of this magnitude to have an influence on the high cloud area identified. However, we have chosen a cloud fraction of 1 for clarity - any other fraction is arbitrary, and makes it less apparent how to define CTT of such cells. In addition, particularly thin parts (i.e. low cloud fraction or mass parts) of anvil cloud are unlikely to be a major contribution to anvil radiative effect, as shown in figure 10c of https://doi.org/10.5194/egusphere-2025-203 , and therefore are of less interest to this study.*

*Satellites don't have cloud mass mixing ratio measurements, they only have radiances on which to base their assessment of the cloud mask. They relate retrieval radiances to that expected based on estimates of skin temperature and profiles for clear-sky. Despite the different approaches used between model and satellite to define high cloud, we find that the model estimate of high cloud area performs well (as discussed in Section 4).*

4.Lines 285-286: While the focus may be on how SW CRE varies, this SW bias is often over 20% (from fig 4a). Some discussion should be included regarding how this bias could impact convective formation and diagnosis of SW albedo. Given that this study focuses on radiative effects, this bias is significant.

*We acknowledge that this bias could have an effect on the simulation of convection and high cloud albedo. However, since convective cloud seems to have formed reasonably well in the simulations, and the direct evaluation of high cloud albedo shows good skill (r=0.74) once only one outlier case is disregarded, we are led to conclude that this bias is not of first-order importance for our conclusions. We have added the following text to ensure the reader is aware of the potential influence of the bias:*

*"The subsequent discussion of high cloud specific metrics demonstrates that the mean all-sky SW bias is not a direct result of biases in high cloud radiative properties. Therefore, any bias in the model of consequence to this study would be indirect, e.g. through reduced surface heating and therefore convection. However, it is not apparent that the SW bias has inhibited the model convection, nor has it had a clear detrimental effect on the high cloud specific metrics, discussed below. Understanding the source of this bias is an open area of investigation by relevant modelling groups and, whilst it remains a caveat for this work, we do not believe it has a major influence on conclusions." (L302-308)*

5.Line 291: The verbiage here is mixing the meanings of "lower" and "high". Please refer to altitude using low and high. I assume that "lower" means less in this context. So perhaps use "less" and "more" or "increased" and "decreased" to refer to change in magnitude.

*Thank you, yes, the wording was a bit awkward. We've changed to "reduced high cloud coverage". (L313)*

6.Lines 403-405: Why is the decrease in droplet number more impactful than the increase in droplet number toward change in cloud albedo?

*This is a saturation effect. In a polluted environment, as studied here, incremental increases may have a noticeably reduced effect. Saturation may occur through a radiative mechanism; analogous to liquid clouds (Grosvenor et al., 2017, Appendix C), one could expect optical thickness to vary with $\sim N_{ice}^{(1/3)}$, with $N_{ice}$ having a diminishing impact as it is increased.*

*We have added a sentence on this:*
*"This saturation effect may stem from a radiative pathway, e.g. decreasing strength of the optical depth response to further increasing ice number concentration" (L437)*

*Grosvenor, D. P. et al. (2017). The relative importance of macrophysical and cloud albedo changes for aerosol-induced radiative effects in closed-cell stratocumulus: insight from the modelling of a case study. Atmospheric Chemistry and Physics.*
*https://doi.org/10.5194/acp-17-5155-2017*

7.Lines 453-455: The fact that the INP experiments lead to more ice hydrometeor mass than the increase in droplet experiment, combined with only a small increase in ice number for the

INP experiments compared to the cloud droplet increase experiment, seems counter intuitive to me. Homogeneous freezing of droplets often dominates in adding the most mass and number to convective anvils. The results suggest that the ice hydrometeors are significantly larger in the INP experiments compared to the increased droplet experiment. Why would this be the case? Is there a default ice crystal size (from INP heterogeneous nucleation) in the microphysics scheme that could be influencing this outcome?

Heterogeneously-frozen ice and secondary ice that form at much warmer temperatures than homogeneous freezing are able to then grow during their ascent for several kilometres before they reach the anvil. Since diffusional growth of ice hydrometeors is proportional to the integrated sum of size and number, if growth is not vapour limited, then increasing ice number concentration will lead to faster mass growth of the ice hydrometeors. However, once the ice hydrometeors are formed we no longer know what the origin of them was - homogeneous, heterogeneous or secondary production, and it cannot be quantitatively determined which population is dominating the mass growth.

There is not a constant ice crystal size for heterogeneous ice nucleation – it will be dependent on the ice mass frozen and INP.

The reviewer's comment has drawn us to consider the average ice particle size, and also note that we have not presented results that allowed robust conclusions on this point. Two new panels have been added to Figure 8, and equivalent supplementary figures for snow and ice crystals separately, showing the average particle mass (see new fig below), which is the division of the case profiles that go into the first row of plots by the case profiles in the second row. This now allows us to robustly comment on particle size. The conclusions have also been integrated into the main text and figure 9 diagram.

The diagram (below, and new fig 9) shows that for increased droplet number, there is an increase in anvil ice crystals with no significant change in anvil ice mass, and therefore that ice mass is divided among more ice particles and the average mass per particle decreases. For an increase in INP, both number and mass of ice particles increases. The average mass per particle decreases, illustrating that the mass has not increased proportionally to increases in number of particles. Much of the significant effects occur in the mixed-phase region but there significant effects as cold as 230K.

We have added the following text:
"Average particle mass reduces with increased droplet number, consistent with an increase in ice particle number but no significant increase in ice mass (Figure 8e-f). Increasing INP also reduces average particle mass, highlighting that, in these model experiments, INP can increase the frozen mass in the mixed-phase and lower anvil, but does not do so proportionally to the increase in ice number. Separate profiles of snow and ice crystals average particle mass show that snow profiles are most similar to the combined profile with equivalent responses throughout the vertical profile (Supplementary Figure S11). Ice crystal changes occur in the respective regions of dominant ice formation, i.e. INP perturbations primarily affect average particle mass in the mixed-phase, heterogeneous freezing region, whilst droplet numbers influence the average particle mass in the colder, homogeneous freezing regions of the profile (Supplementary Figure S10)." (L508-515)

We have corrected a statement which made an incorrect inference about the change in particle mass in the INP experiments. This now reads:

"The number and mass of snow hydrometeors increases in the lower part of the thick cloud in this experiment along with a corresponding decrease in average particle mass in that region, so we increase the number of ice hydrometeors in the diagram but decrease their size." (L537-539)

[Figure]

[Figure]

**Mean albedo of *both* thick and thin high cloud increased, by increased ice crystal number in both cloud regions**

*Control*

Thick high cloud

Thin high cloud

*High droplet number*

**High cloud albedo increased by a microphysical perturbation**

*High ice nucleating particles*

**Mean albedo of *only* thick high cloud increased, by increased snow number and mass in the mid to lower anvil**